# INVARIANT CAUSAL REPRESENTATION LEARNING FOR OUT-OF-DISTRIBUTION GENERALIZATION

**Chaochao Lu**[1,2], **Yuhuai Wu**[3,4†], **Jose Miguel Hernández-Lobato**[1,5*], **Bernhard Schölkopf** [2*]

## ABSTRACT

Due to spurious correlations, machine learning systems often fail to generalize to environments whose distributions differ from the ones used at training time. Prior work addressing this, either explicitly or implicitly, attempted to find a data representation that has an invariant relationship with the target. This is done by leveraging a diverse set of training environments to reduce the effect of spurious features and build an invariant predictor. However, these methods have generalization guarantees only when both data representation and classifiers come from a linear model class. We propose invariant Causal Representation Learning (iCaRL), an approach that enables out-of-distribution (OOD) generalization in the nonlinear setting (i.e., nonlinear representations and nonlinear classifiers). It builds upon a practical and general assumption: the prior over the data representation (i.e., a set of latent variables encoding the data) given the target and the environment belongs to general exponential family distributions, i.e., a more flexible conditionally non-factorized prior that can actually capture complicated dependences between the latent variables. Based on this, we show that it is possible to identify the data representation up to simple transformations. We also show that all direct causes of the target can be fully discovered, which further enables us to obtain generalization guarantees in the nonlinear setting. Experiments on both synthetic and real-world datasets demonstrate that our approach outperforms a variety of baseline methods.

## 1 INTRODUCTION

Modern machine learning algorithms still lack robustness, and may fail to generalize outside of a specific training distribution because they learn easy-to-fit spurious correlations which are prone to change between training and testing environments. We recall the widely used example of classifying images of camels and cows (Beery et al., 2018). Here, the training dataset has a selection bias, i.e., many pictures of cows are taken on green pastures, while most pictures of camels happen to be in deserts. After training, it is found that the model builds on spurious correlations, i.e., it relates green pastures with cows and deserts with camels, and fails to recognize images of cows on the beach.

To address this problem, a natural idea is to identify which features of the training data present domain-varying spurious correlations with labels and which features describe true correlations of interest that are stable across domains. In the example above, the former are the features describing the context (e.g., pastures and deserts), whilst the latter are the features describing the animals (e.g., animal shape). By exploiting the varying degrees of spurious correlation naturally present in training data collected from multiple environments, one can try to identify stable features and build invariant predictors. Invariant risk minimization (IRM) seeks to find data representations (Arjovsky et al., 2019) or features (Rojas-Carulla et al., 2018) for which the optimal predictor is invariant across all environments. The general formulation of IRM is a challenging bi-leveled optimization problem, and theoretical guarantees require constraining both data representations and classifiers to be linear (Arjovsky et al., 2019, Theorem 9), or considering the special case of feature selection (Rojas-Carulla et al., 2018, Theorem 4). Ahuja et al. (2020a) study the problem from the perspective of game theory, with an approach termed invariant risk minimization games (IRMG). They show that the set of Nash equilibria for a proposed game is equivalent to the set of invariant predictors for any finite number of environments, even with nonlinear data representations and nonlinear classifiers. However, these

---

[1]University of Cambridge, [2]MPI for Intelligent Systems, [3]Stanford University, [4]Google Research, [5]The Alan Turing Institute, [†]Work done at University of Toronto, [*]Equal Supervision, Correspondence at cl641@cam.ac.uk.

theoretical results in the nonlinear setting only guarantee that one can learn invariant predictors from training environments, but do not guarantee that the learned invariant predictors can generalize well across all environments including unseen testing environments.

We propose invariant Causal Representation Learning (iCaRL), a novel approach that enables out-of-distribution (OOD) generalization in the nonlinear setting (i.e., nonlinear representations and nonlinear classifiers[1]). We achieve this by extending and using methods from representation learning and graphical causal discovery. In more detail, we first introduce our main *general assumption*: when conditioning on the target (e.g., labels) and the environment (represented as an index), the prior over the data representation (i.e., a set of latent variables encoding the data) belongs to a *general exponential family*. Unlike the conditionally factorized prior assumed in recent identifiable variational autoencoders (iVAE) (Khemakhem et al., 2020a), this is a more flexible conditionally non-factorized prior, which can actually capture complicated dependences between the latent variables. We then extend iVAE to the case in which the latent variable prior belongs to such a *general exponential family*. The combination of this result and the previous general assumption allows us to guarantee that the data representation can be identified up to simple transformations. We then show that the direct causes of the target can be fully discovered by analyzing all possible graphs in a structural equation model setting. Once they are discovered, the challenging bi-leveled optimization problem in IRM and IRMG can be reduced to two simpler independent optimization problems, that is, learning the data representation and learning the optimal classifier can be performed separately. This leads to a practical algorithm and enables us to obtain generalization guarantees in the nonlinear setting.

Overall, we make a number of key contributions: (1) We propose a general framework for out-of-distribution generalization in the nonlinear setting with the theoretical guarantees on both identifiability and generalizability; (2) We propose a general assumption on the underlying causal diagram for prediction (Assumption 1 and Fig. 1c), which covers many real-world scenarios (Section 3.2); (3) We propose a general assumption on the prior over the latent variables (Assumption 2), i.e., a more flexible conditionally non-factorized prior; (4) We prove that an extended iVAE with this conditionally non-factorized prior is also identifiable (Theorems 1, 2&3); (5) We prove that our framework has the theoretical guarantees for OOD generalization in the nonlinear setting (Proposition 1).

## 2 PRELIMINARIES

### 2.1 IDENTIFIABLE VARIATIONAL AUTOENCODERS

Variational autoencoders (VAEs, see Appendix B) (Kingma & Welling, 2013; Rezende et al., 2014) lack identifiability guarantees. Consider a VAE model where $X \in \mathbb{R}^d$ stands for the observed variables (data) and $Z \in \mathbb{R}^n$ for the latent variables. Khemakhem et al. (2020a) show that a VAE with an unconditional prior distribution $p_\theta(Z)$ over the latent variables is unidentifiable. However, they also show that it is possible to obtain an identifiable model if one posits a conditionally factorized prior distribution over the latent variables, $p_\theta(Z|U)$, where $U \in \mathbb{R}^m$ is an additional observed variable (Hyvärinen et al., 2019). Specifically, let $\theta = (f, T, \lambda) \in \Theta$ be the parameters of the conditional generative model

$$p_\theta(X, Z|U) = p_f(X|Z)p_{T,\lambda}(Z|U),  \tag{1}$$

where $p_f(X|Z) = p_\epsilon(X - f(Z))$ in which $\epsilon$ is an independent noise variable with probability density function $p_\epsilon(\epsilon)$. Importantly, the prior $p_{T,\lambda}(Z|U)$ is assumed to be conditionally factorial, where each element of $Z_i \in Z$ has a univariate exponential family distribution given $U$. The conditioning on $U$ is through an arbitrary function $\lambda(U)$ (e.g., a neural net) that outputs the individual exponential family parameters $\lambda_i(U)$ for each $Z_i$. The prior probability density thus takes the form

$$p_{T,\lambda}(Z|U) = \prod_i \mathcal{Q}_i(Z_i)/\mathcal{C}_i(U) \exp\left[\sum_{j=1}^k T_{i,j}(Z_i)\lambda_{i,j}(U)\right],  \tag{2}$$

where $\mathcal{Q}_i$ is the base measure, $Z_i$ the $i$-th dimension of $Z$, $\mathcal{C}_i(U)$ the normalizing constant, $T_i = (T_{i,1}, \ldots, T_{i,k})$ the sufficient statistics, $\lambda_i(U) = (\lambda_{i,1}(U), \ldots, \lambda_{i,k}(U))$ the corresponding natural parameters depending on $U$, and $k$ the dimension of each sufficient statistic that is fixed in advance. It is worth noting that this prior is restrictive as it is factorial and therefore cannot capture dependencies. As in VAEs, the model parameters are estimated by maximizing the corresponding evidence lower bound (ELBO),

$$\mathcal{L}_{\text{iVAE}}(\theta, \phi) := \mathbb{E}_{p_D}\left[\mathbb{E}_{q_\phi(Z|X,U)}\left[\log p_f(X|Z) + \log p_{T,\lambda}(Z|U) - \log q_\phi(Z|X,U)\right]\right],  \tag{3}$$

---

[1]In fact, we are not restricted to the classification case and allow the target to be either continuous or categorical, which will be formally defined in Section 2.2.

where we denote by $p_D$ the empirical data distribution given by the dataset $\mathcal{D} = \left\{ \left( \boldsymbol{X}^{(i)}, \boldsymbol{U}^{(i)} \right) \right\}_{i=1}^{N}$ and $q_{\boldsymbol{\phi}}(\boldsymbol{Z}|\boldsymbol{X}, \boldsymbol{U})$ denotes an approximate conditional distribution for $\boldsymbol{Z}$ given by a recognition network with parameters $\boldsymbol{\phi}$. This approach is called identifiable VAE (iVAE). Most importantly, it can be proved that under the conditions stated in Theorem 2 of Khemakhem et al. (2020a), iVAE can identify the latent variables $\boldsymbol{Z}$ up to a permutation and a simple componentwise transformation, see Appendix F.

## 2.2 INVARIANT RISK MINIMIZATION

Arjovsky et al. (2019) introduced invariant risk minimization (IRM), whose goal is to construct an *invariant predictor* $f$ that performs well across all environments $\mathcal{E}_{all}$ by exploiting data collected from multiple environments $\mathcal{E}_{tr}$, where $\mathcal{E}_{tr} \subseteq \mathcal{E}_{all}$. Technically, they consider datasets $D_e := \{(\boldsymbol{x}_i^e, \boldsymbol{y}_i^e)\}_{i=1}^{n_e}$ from multiple training environments $e \in \mathcal{E}_{tr}$, where $\boldsymbol{x}_i^e \in \mathcal{X} \subseteq \mathbb{R}^d$ is the input observation and its corresponding label is $\boldsymbol{y}_i^e \in \mathcal{Y} \subseteq \mathbb{R}^s$.[2] The dataset $D_e$, collected from environment $e$, consists of examples identically and independently distributed according to some probability distribution $P(\boldsymbol{X}^e, \boldsymbol{Y}^e)$. The goal of IRM is to use these multiple datasets to learn a predictor $\boldsymbol{Y} = f(\boldsymbol{X})$ that performs well for all the environments. Here we define the risk reached by $f$ in environment $e$ as $R^e(f) = \mathbb{E}_{\boldsymbol{X}^e, \boldsymbol{Y}^e} [\ell(f(\boldsymbol{X}^e), \boldsymbol{Y}^e)]$, where $\ell(\cdot)$ is a loss function. Then, the invariant predictor can be formally defined as follows:

**Definition 1** (Invariant Predictor (Arjovsky et al., 2019))**.** *We say that a data representation* $\Phi \in \mathcal{H}_{\Phi} : \mathcal{X} \to \mathcal{F}$ *elicits an invariant predictor* $w \circ \Phi$ *across environments* $\mathcal{E}$ *if there is a classifier* $w \in \mathcal{H}_w : \mathcal{F} \to \mathcal{Y}$ *simultaneously optimal for all environments, that is,* $w \in \arg\min_{\bar{w} \in \mathcal{H}_w} R^e(\bar{w} \circ \Phi)$ *for all* $e \in \mathcal{E}$, *where* $\circ$ *means function composition.*

Mathematically, IRM can be phrased as the following constrained optimization problem:

$$\min_{\Phi \in \mathcal{H}_\Phi, w \in \mathcal{H}_w} \sum_{e \in \mathcal{E}_{tr}} R^e(w \circ \Phi) \quad \text{s.t. } w \in \arg\min_{\bar{w} \in \mathcal{H}_w} R^e(\bar{w} \circ \Phi), \forall e \in \mathcal{E}_{tr}. \quad (4)$$

Since this is a generally infeasible bi-leveled optimization problem, Arjovsky et al. (2019) rephrased it as a tractable penalized optimization problem by transfering the inner optimization routine to a penalty term. The main generalization result (Theorem 9 in Arjovsky et al. (2019)) states that if both $\Phi$ and $w$ come from the class of linear models (i.e., $\mathcal{H}_\Phi = \mathbb{R}^{n \times n}$ and $\mathcal{H}_w = \mathbb{R}^{n \times 1}$), under certain conditions on the diversity of training environments (Assumption 8 in Arjovsky et al. (2019)) and the data generation, the invariant predictor $w \circ \Phi$ obtained by solving Eq. (4) remains invariant in $\mathcal{E}_{all}$.

## 3 PROBLEM SETUP

### 3.1 A MOTIVATING EXAMPLE

In this section, we extend the example which was introduced by Wright (1921) and discussed by Arjovsky et al. (2019), and provide a further in-depth analysis.

**Model 1.** *Consider a structural equation model (SEM) with a discrete environment variable $E$ that modulates the noises in the structural assignments connecting the other variables (cf. Fig. 1a below):*
$$Z_1 \leftarrow Gaussian(0, \sigma_1(E)), \quad Y \leftarrow Z_1 + Gaussian(0, \sigma_2(E)), \quad Z_2 \leftarrow Y + Gaussian(0, \sigma_3(E)),$$
*where $Gaussian(0, \sigma)$ denotes a Gaussian random variable with zero mean and standard deviation $\sigma$, and $\sigma_1, \ldots, \sigma_3$ are functions of the value $e \in \mathcal{E}_{all}$ taken by the environment variable $E$.*

To ease exposition, here we consider the simple scenario in which $\mathcal{E}_{all}$ only contains all modifications varying the noises of $Z_1$, $Z_2$ and $Y$ within a finite range, i.e., $\sigma_i(e) \in [0, \sigma_{max}^2]$. Then, to predict $Y$ from $(Z_1, Z_2)$ using a least-square predictor $\hat{Y}^e = \hat{\alpha}_1 Z_1^e + \hat{\alpha}_2 Z_2^e$ for environment $e$, we can

- Case 1: regress from $Z_1^e$, to obtain $\hat{\alpha}_1 = 1$ and $\hat{\alpha}_2 = 0$,
- Case 2: regress from $Z_2^e$, to obtain $\hat{\alpha}_1 = 0$ and $\hat{\alpha}_2 = \frac{\sigma_1(e) + \sigma_2(e)}{\sigma_1(e) + \sigma_2(e) + \sigma_3(e)}$,
- Case 3: regress from $(Z_1^e, Z_2^e)$, to obtain $\hat{\alpha}_1 = \frac{\sigma_3(e)}{\sigma_2(e) + \sigma_3(e)}$ and $\hat{\alpha}_2 = \frac{\sigma_2(e)}{\sigma_2(e) + \sigma_3(e)}$.

In the generic scenario (i.e., $\sigma_1(e) \neq 0$, $\sigma_2(e) \neq 0$, and $\sigma_3(e) \neq 0$), the regression using $Z_1$ in Case 1 is an invariant correlation: it is the only regression whose coefficients do not vary with $e$. By contrast,

---

[2]The setup applies to both continuous and categorical data. If any observation or label is categorical, we could one-hot encode it.

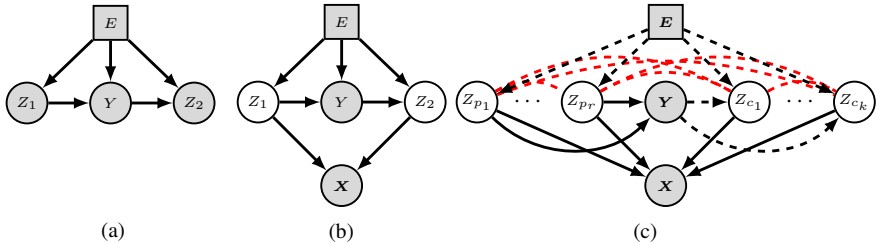

Figure 1: (a) Causal structure of Model 1. (b) A more practical extension of Model 1, where $Z_1$ and $Z_2$ are not directly observed and $X$ is their observation. (c) A general version of (b), where we assume there exist multiple unobserved variables. Each of them could be either a parent, a child of $Y$, or has no direct connection with $Y$. We allow for arbitrary connections between the latent variables (red dashed lines) as long as the resulting causal diagram including $Y$ is a directed acyclic graph (DAG). Grey nodes denote observed variables and white nodes represent unobserved variables. Dashed lines denote the edges which might vary across environments and even be absent in some scenarios, whilst solid lines indicate that they are invariant across all the environments.

the regressions in both Case 2 and Case 3 have coefficients that depend on $e$. Therefore, only the invariant correlation in Case 1 will generalize well to new test environments.

Another way to understand Model 1 is through its graphical representation[3], as shown in Fig. 1a. We treat the environment as a random variable $E$, where $E$ could be any information specific to the environment (Storkey, 2009; Peters et al., 2015; Zhang et al., 2017; Huang et al., 2020). Unless stated otherwise, for simplicity, we let $E$ be the environment index, i.e., $E \in \{1, \ldots, N\}$, where $N$ is the number of training environments. A more realistic version appearing in many settings is shown in Fig. 1b, where the true variables $\{Z_1, Z_2\}$ are unobserved and we can only observe their transformation $X$. In this case, Invariant Causal Prediction (ICP) (Peters et al., 2015) will fail when applied to $X$, even when $Y$ is not affected by $E$ (i.e., the edge $E \to Y$ is removed). The reason is that each variable (i.e., each dimension) of $X$ is jointly influenced by both $Z_1$ and $Z_2$ so that ICP is unable to find the variables containing the information only about $Z_1$ by searching for a subset of variables $X$. By contrast, both IRM and IRMG work, as long as the transformation is linear. These findings are also empirically illustrated in Appendix K.1. We now go even further and consider a more general causal graph in which $Y$ can have more than one parent or child.

## 3.2 Assumptions on the Causal Graph

We extend the causal graph in Fig. 1b to a more general setting[4], as encapsulated in Fig. 1c. In particular, we now have $X \in \mathcal{X} \subseteq \mathbb{R}^d$, $Y \in \mathcal{Y} \subseteq \mathbb{R}^s$, $Z = (Z_{p_1}, \ldots, Z_{p_r}, Z_{c_1}, \ldots, Z_{c_k}) \in \mathcal{Z} \subseteq \mathbb{R}^n$, where $n = r + k$, and $\{Z_i\}_{i \in I_p = \{p_1, \ldots, p_r\}}$ and $\{Z_j\}_{j \in I_c = \{c_1, \ldots, c_k\}}$ are multiple scalar *causal factors* and *non-causal factors*[5] of $Y$, respectively. We denote $Z_p \doteq (Z_{p_1}, \ldots, Z_{p_r})$ and $Z_c \doteq (Z_{c_1}, \ldots, Z_{c_k})$ for the ease of clarification. We also assume that $Z$ is of lower dimension than $X$, that is, $n \leq d$. We allow for arbitrary connections between the latent variables $Z$ as long as the resulting causal diagram including $Y$ is a directed acyclic graph (DAG). We use dashed lines to indicate the *causal mechanisms* which might vary across environments and even be absent in some scenarios, whilst solid lines indicate that they are invariant across all the environments. To sum up, we assume that the underlying causal graph encapsulated in Fig. 1c satisfies the following assumption[6]:

**Assumption 1.** *(a) $Z_i$ depends on one or both of $Y$ and $E$ for any $i$; (b) The causal graph containing $Z$ and $Y$ is a DAG; (c) $X \perp\!\!\!\perp Y, E | Z$, implying that $p(X|Z)$ is invariant across all the environments; (d) $Y \perp\!\!\!\perp E | Z_p$, implying that $p(Y|Z_p)$ is invariant across all the environments.*

One may be wondering how practical Assumption 1 is in real world applications. Let us explore this in more detail. Assumption 1a rules out all the useless $Z_i$ in the task of predicting $Y$. This is because if Assumption 1a is violated, meaning that $Z_i$ is independent of $Y$ and $E$ and has no influence in predicting $Y$, then such $Z_i$ should be viewed as noise and thus eliminated during learning. Assumption 1b is a common assumption in causal discovery (Spirtes et al., 2000; Pearl, 2009; Peters et al., 2015). It also makes sense in Assumption 1c that the generative mechanism $p(X|Z)$ is

---

[3]The relation between SEM and its graphical representation is formally defined in Appendix D.

[4]For simplicity, we do not explicitly consider unobserved confounders in this paper. In particular, we assume that there are no unobserved confounders between $Z, Y, X$, and $E$.

[5]This means that $Z_{j \in I_c}$ could be either an effect of $Y$, independent of $Y$, or spuriously correlated with $Y$ via a third set of confounders (i.e., both $Z_j$ and $Y$ are affected by a subset of $\{Z_i\}_{i \neq j}$ and $E$).

[6]For generality, we replace $E$ with $E$ to additionally include the case of multi-dimensional variables.

---

**Algorithm 1:** Invariant Causal Representation Learning (iCaRL)

---

**Phase 1:** We first learn a NF-iVAE model, including the decoder and its corresponding encoder, by optimizing the objective function in (10) on the data $\{\boldsymbol{X}, \boldsymbol{Y}, \boldsymbol{E}\}$. Then, we use the mean of the NF-iVAE encoder to infer the latent variables $\boldsymbol{Z}$ from observations $\{\boldsymbol{X}, \boldsymbol{Y}, \boldsymbol{E}\}$. The latent variables are guaranteed to be identified up to a permutation and simple transformation.

**Phase 2:** After inferring $\boldsymbol{Z}$, we first conduct the PC algorithm to learn a Markov equivalence class of DAGs, and then discover direct causes (parents) of $\boldsymbol{Y}$ among its neighbors by testing all pairs of latent variables with (conditional) independence testing, i.e., finding a set of latent variables in which each pair of $Z_i$ and $Z_j$ satisfies that the dependency between them increases after additionally conditioning on $\boldsymbol{Y}$.

**Phase 3:** Having obtained Pa($\boldsymbol{Y}$), we can solve (11) to learn the invariant classifier $w$. When in a new environment, we first infer Pa($\boldsymbol{Y}$) from $\boldsymbol{X}$ by solving (12) and then leverage the learned $w$ for prediction.

---

invariant across all the environments. Otherwise, it is impossible to infer $\boldsymbol{Z}$ from $\boldsymbol{X}$ in any unseen environment. Assumption 1d is a widely-used default assumption in OOD generalization (Peters et al., 2015; Arjovsky et al., 2019). In fact, Assumption 1d can be further relaxed to the more practical one that $\mathbb{E}[\boldsymbol{Y}|\boldsymbol{Z}_p]$ is invariant across all the environments. That is, given $\boldsymbol{Z}_p$, we allow $\boldsymbol{E}$ to only affect the amount of noise in the distribution of $\boldsymbol{Y}$, because that would not change the expected value of $\boldsymbol{Y}$ since the mean of the noise would be zero. Apparently, Assumption 1, together with the causal graph in Fig. 1c, covers most scenarios (e.g., the ones of Zhang et al. (2013); von Kügelgen et al. (2020); Sun et al. (2020); Ahuja et al. (2021); von Kügelgen et al. (2021)) and is a very flexible model for causal analysis when predicting $\boldsymbol{Y}$ from $\boldsymbol{X}$.

### 3.3 ASSUMPTIONS ON THE PRIOR

When the underlying causal graph satisfies Assumption 1, our primary assumption leading to identifiability in this general setting is that the conditional prior $p(\boldsymbol{Z}|\boldsymbol{Y}, \boldsymbol{E})$ belongs to a general exponential family. This is formalized as follows:

**Assumption 2.** $p(\boldsymbol{Z}|\boldsymbol{Y}, \boldsymbol{E})$ *belongs to a general exponential family with parameter vector given by an arbitrary function* $\boldsymbol{\lambda}(\boldsymbol{Y}, \boldsymbol{E})$ *and sufficient statistics* $\boldsymbol{T}(\boldsymbol{Z}) = [\boldsymbol{T}_f(\boldsymbol{Z})^T, \boldsymbol{T}_{NN}(\boldsymbol{Z})^T]^T$ *given by the concatenation of a) the sufficient statistics* $\boldsymbol{T}_f(\boldsymbol{Z}) = [\boldsymbol{T}_1(Z_1)^T, \cdots, \boldsymbol{T}_n(Z_n)^T]^T$ *of a factorized exponential family, where all the* $\boldsymbol{T}_i(Z_i)$ *have dimension larger or equal to 2, and b) the output* $\boldsymbol{T}_{NN}(\boldsymbol{Z})$ *of a neural network with ReLU activations. The resulting density function is thus given by*

$$p_{\boldsymbol{T},\boldsymbol{\lambda}}(\boldsymbol{Z}|\boldsymbol{Y}, \boldsymbol{E}) = \mathcal{Q}(\boldsymbol{Z})/\mathcal{C}(\boldsymbol{Y}, \boldsymbol{E})\exp\left[\boldsymbol{T}(\boldsymbol{Z})^T\boldsymbol{\lambda}(\boldsymbol{Y}, \boldsymbol{E})\right], \quad (5)$$

*where* $\mathcal{Q}$ *is the base measure and* $\mathcal{C}$ *the normalizing constant.*

A neural network with ReLU activation has universal approximation power. Therefore, the term $\boldsymbol{T}_{NN}(\boldsymbol{Z})$ in the above prior distribution will allow us to capture arbitrary dependencies between the latent variables. The distribution in Eq. (5) is more flexible than the conditionally factorized prior assumed by iVAEs. However, surprisingly, the identifiability results of iVAEs also hold when using the more flexible prior in Eq. (5), as we will show in Section 4.1. This motivates using an extended iVAE model with the above prior to model data generated by the ground truth model in Fig. 1c. However, in this case, the data generating model and the learned model might have different priors. For example, in the ground truth model, the prior for each $Z_{i \in I_p}$ might be $p(Z_i|\boldsymbol{E})$, when $Z_i$ is only caused by $\boldsymbol{E}$. By contrast, in the extended iVAE model the prior is $p(\boldsymbol{Z}|\boldsymbol{Y}, \boldsymbol{E})$. Is this going to affect the identifiability of the latent variables? Well, in practice not because the posterior distribution for $\boldsymbol{Z}$ given $\boldsymbol{X}$, $\boldsymbol{Y}$ and $\boldsymbol{E}$ would be equivalent in both models (up to identifiability guarantees).

## 4 INVARIANT CAUSAL REPRESENTATION LEARNING

We now introduce our algorithm, invariant Causal Representation Learning (iCaRL), which consists of 3 phases as summarized in Algorithm 1. The idea is that we first identify $\boldsymbol{Z}$ by using an extended iVAE model under Assumptions 1&2 (Phase 1), then discover direct causes of $\boldsymbol{Y}$ among the identified $\boldsymbol{Z}$ (Phase 2), and finally learn an invariant predictor for $\boldsymbol{Y}$ from the discovered causes (Phase 3).

### 4.1 PHASE 1: IDENTIFYING LATENT VARIABLES USING NF-IVAE

In this section, we describe our identifiable model, namely NF-iVAE, which is an extended iVAE with a general non-factorized prior that is able to capture complex dependences between the latent variables. Technically, in the general setting under Assumption 1, it is straightforward to obtain a

corresponding generative model by directly substituting $\boldsymbol{U}$ with $(\boldsymbol{Y}, \boldsymbol{E})$ in Eq. (1):

$$p_{\boldsymbol{\theta}}(\boldsymbol{X}, \boldsymbol{Z}|\boldsymbol{Y}, \boldsymbol{E}) = p_{\boldsymbol{f}}(\boldsymbol{X}|\boldsymbol{Z})p_{\boldsymbol{T}, \boldsymbol{\lambda}}(\boldsymbol{Z}|\boldsymbol{Y}, \boldsymbol{E}), \tag{6}$$

$$p_{\boldsymbol{f}}(\boldsymbol{X}|\boldsymbol{Z}) = p_{\boldsymbol{\epsilon}}(\boldsymbol{X} - \boldsymbol{f}(\boldsymbol{Z})). \tag{7}$$

The corresponding ELBO is

$$\mathcal{L}_{\text{phase1}}^{\text{ELBO}}(\boldsymbol{\theta}, \boldsymbol{\phi}) := \mathbb{E}_{p_D}\left[\mathbb{E}_{q_{\boldsymbol{\phi}}(\boldsymbol{Z}|\boldsymbol{X}, \boldsymbol{Y}, \boldsymbol{E})}\left[\log p_{\boldsymbol{f}}(\boldsymbol{X}|\boldsymbol{Z}) + \log p_{\boldsymbol{T}, \boldsymbol{\lambda}}(\boldsymbol{Z}|\boldsymbol{Y}, \boldsymbol{E}) - \log q_{\boldsymbol{\phi}}(\boldsymbol{Z}|\boldsymbol{X}, \boldsymbol{Y}, \boldsymbol{E})\right]\right]. \tag{8}$$

To obtain an identifiability result, we assume that the prior $p_{\boldsymbol{T}, \boldsymbol{\lambda}}(\boldsymbol{Z}|\boldsymbol{Y}, \boldsymbol{E})$ satisfies Assumption 2 (i.e., Eq. (5)). Since the prior is a general multivariate exponential family distribution with unknown normalization constant, we cannot learn its parameters $(\boldsymbol{T}, \boldsymbol{\lambda})$ by directly maximizing Eq. (8). Instead, we use score matching, a well-known method for training unnormalized probabilistic models (Hyvärinen, 2005; Vincent, 2011), and learn $(\boldsymbol{T}, \boldsymbol{\lambda})$ by minimizing

$$\mathcal{L}_{\text{phase1}}^{\text{SM}}(\boldsymbol{T}, \boldsymbol{\lambda}) := \mathbb{E}_{p_D}\left[\mathbb{E}_{q_{\boldsymbol{\phi}}(\boldsymbol{Z}|\boldsymbol{X}, \boldsymbol{Y}, \boldsymbol{E})}\left[||\nabla_{\boldsymbol{Z}} \log q_{\boldsymbol{\phi}}(\boldsymbol{Z}|\boldsymbol{X}, \boldsymbol{Y}, \boldsymbol{E}) - \nabla_{\boldsymbol{Z}} \log p_{\boldsymbol{T}, \boldsymbol{\lambda}}(\boldsymbol{Z}|\boldsymbol{Y}, \boldsymbol{E})||^2\right]\right]. \tag{9}$$

In practice, we can use a simple trick of partial integration to simplify the evaluation of Eq. (9), see Appendix C. Furthermore, we can jointly learn $(\boldsymbol{\theta}, \boldsymbol{\phi})$ by combining Eq. (8) and Eq. (9) in the following objective:

$$\mathcal{L}_{\text{phase1}}(\boldsymbol{\theta}, \boldsymbol{\phi}) = \mathcal{L}_{\text{phase1}}^{\text{ELBO}}(\boldsymbol{f}, \hat{\boldsymbol{T}}, \hat{\boldsymbol{\lambda}}, \boldsymbol{\phi}) - \mathcal{L}_{\text{phase1}}^{\text{SM}}(\hat{\boldsymbol{f}}, \boldsymbol{T}, \boldsymbol{\lambda}, \hat{\boldsymbol{\phi}}), \tag{10}$$

where $\hat{\boldsymbol{f}}, \hat{\boldsymbol{T}}, \hat{\boldsymbol{\lambda}}, \hat{\boldsymbol{\phi}}$ are copies of $\boldsymbol{f}, \boldsymbol{T}, \boldsymbol{\lambda}, \boldsymbol{\phi}$ that are treated as constants and whose gradient is not calculated during learning. More details can be found in Appendix M.

We now state our main theoretical results:

**Theorem 1.** *Assume that we observe data sampled from a generative model defined according to Eqs. (5-7), with parameters $\boldsymbol{\theta} := (\boldsymbol{f}, \boldsymbol{T}, \boldsymbol{\lambda})$, where $p_{\boldsymbol{T}, \boldsymbol{\lambda}}(\boldsymbol{Z}|\boldsymbol{Y}, \boldsymbol{E})$ satisfies Assumption 2. Furthermore, assume the following holds: (i) The set $\{\boldsymbol{X} \in \mathcal{O}|\varphi_{\boldsymbol{\epsilon}}(\boldsymbol{X}) = 0\}$ has measure zero, where $\varphi_{\boldsymbol{\epsilon}}$ is the characteristic function of the density $p_{\boldsymbol{\epsilon}}$ defined in Eq. (7). (ii) Function $\boldsymbol{f}$ in Eq. (7) is injective, and has all second-order cross derivatives. (iii) The sufficient statistics in $\boldsymbol{T}_f$ are all twice differentiable. (iv) There exist $k + 1$ distinct points $(\boldsymbol{Y}, \boldsymbol{E})^0, \ldots, (\boldsymbol{Y}, \boldsymbol{E})^k$ such that the matrix $L = \left(\boldsymbol{\lambda}((\boldsymbol{Y}, \boldsymbol{E})^1) - \boldsymbol{\lambda}((\boldsymbol{Y}, \boldsymbol{E})^0), \ldots, \boldsymbol{\lambda}((\boldsymbol{Y}, \boldsymbol{E})^k) - \boldsymbol{\lambda}((\boldsymbol{Y}, \boldsymbol{E})^0)\right)$ of size $k \times k$ is invertible, where $k$ is the dimension of $\boldsymbol{T}$. Then the parameters $\boldsymbol{\theta}$ are identifiable up to a permutation and a "simple transformation" of the latent variables $\boldsymbol{Z}$, defined as a componentwise nonlinearity making each recovered $\boldsymbol{T}_i(Z_i)$ in $\boldsymbol{T}_f(\boldsymbol{Z})$ equal to the original up to a linear operation.*

Note that, this theorem is inspired by but beyond the main results of iVAEs in that the former is predicated on Assumption 2 which is more flexible than the conditionally factorized prior assumed in iVAEs. It results in several key changes in the proof, clarified in Appendix H. Interestingly, from (iv) we can further see that $\boldsymbol{E}$ is unnecessary when there exist $k + 1$ distinct points $\boldsymbol{Y}^0, \ldots, \boldsymbol{Y}^k$ such that the matrix $L = \left(\boldsymbol{\lambda}(\boldsymbol{Y}^1) - \boldsymbol{\lambda}(\boldsymbol{Y}^0), \ldots, \boldsymbol{\lambda}(\boldsymbol{Y}^k) - \boldsymbol{\lambda}(\boldsymbol{Y}^0)\right)$ of size $k \times k$ is invertible. Not requiring $\boldsymbol{E}$ would make our approach even more applicable.

We further have the following consistency result for the estimation.

**Theorem 2.** *Assume that the following holds: (i) The family of distributions $q_{\boldsymbol{\phi}}(\boldsymbol{Z}|\boldsymbol{X}, \boldsymbol{Y}, \boldsymbol{E})$ contains $p_{\boldsymbol{\theta}}(\boldsymbol{Z}|\boldsymbol{X}, \boldsymbol{Y}, \boldsymbol{E})$, and $q_{\boldsymbol{\phi}}(\boldsymbol{Z}|\boldsymbol{X}, \boldsymbol{Y}, \boldsymbol{E}) > 0$ everywhere. (ii) We maximize $\mathcal{L}_{phase1}(\boldsymbol{\theta}, \boldsymbol{\phi})$ with respect to both $\boldsymbol{\theta}$ and $\boldsymbol{\phi}$. Then in the limit of infinite data, we learn the true parameters $\boldsymbol{\theta}^*$ up to a permutation and simple transformation of the latent variables $\boldsymbol{Z}$.*

As a consequence of Theorems 1&2, we have:

**Theorem 3.** *Assume the hypotheses of Theorem 1 and Theorem 2 hold, then in the limit of infinite data, we identify the true latent variables $\boldsymbol{Z}^*$ up to a permutation and simple transformation.*

Theorem 3 states that we can use NF-iVAE to infer the true $\boldsymbol{Z}^*$ up to a permutation and simple transformation. We use the mean of $q_{\boldsymbol{\phi}}(\boldsymbol{Z}|\boldsymbol{X}, \boldsymbol{Y}, \boldsymbol{E})$ for this task. Note that the noise $\boldsymbol{\epsilon}$ may introduce uncertainty in the estimation of $\boldsymbol{Z}$. However, when $\boldsymbol{X}$ is high dimensional and $\boldsymbol{Z}$ is low dimensional (as common in real world applications), $q_{\boldsymbol{\phi}}(\boldsymbol{Z}|\boldsymbol{X}, \boldsymbol{Y}, \boldsymbol{E})$ will be highly concentrated and we will still be able to estimate $\boldsymbol{Z}$ with high accuracy. The good results obtained by our method in various experiments seem to corroborate this. All three theorems are proven in Appendix H.

### 4.2 PHASE 2: DISCOVERING DIRECT CAUSES

After estimating $\boldsymbol{Z}$ for each data point, the next step is to determine which components of $\boldsymbol{Z}$ are direct causes of $\boldsymbol{Y}$. We denote these components by $\text{Pa}(\boldsymbol{Y})$. We first conduct the PC algorithm (Spirtes et al., 2000) to learn a Markov equivalence class of DAGs, which gives us the direct neighbors of $\boldsymbol{Y}$,

denoted by $\text{Ne}(\boldsymbol{Y})$. Then, from Assumption 1, one observation is that in the generic case, for any two latent variables $Z_i$ and $Z_j$ from $\text{Ne}(\boldsymbol{Y})$, only when both are causes of $\boldsymbol{Y}$ do we have that the dependency between them increases after additionally conditioning on $\boldsymbol{Y}$. Thus, when there exist at least two causal latent variables in $\text{Ne}(\boldsymbol{Y})$, we can test all pairs of latent variables[7] with conditional independence testing[8] (Zhang et al., 2012) to discover $\text{Pa}(\boldsymbol{Y})$ by comparing $p$-values from the two tests: $\texttt{IndTest}(Z_i, Z_j | \boldsymbol{E})$ and $\texttt{IndTest}(Z_i, Z_j | \boldsymbol{Y}, \boldsymbol{E})$, where $\texttt{IndTest}$ denotes (conditional) independence test. Conversely, if no such a pair is found, it implies that there is at most one causal latent variable. This is a highly unlikely case in real world applications, which is left to Appendix G.

### 4.3 Phase 3: Learning an Invariant Predictor

After having obtained the causal latent variables $\text{Pa}(\boldsymbol{Y})$ for $\boldsymbol{Y}$ across training environments, we can learn $w$ by solving the following optimization problem:

$$\min_{w \in \mathcal{H}_w} \sum_{e \in \mathcal{E}_{tr}} R^e(w) = \min_{w \in \mathcal{H}_w} \sum_{e \in \mathcal{E}_{tr}} \mathbb{E}_{\text{Pa}(\boldsymbol{Y}^e), \boldsymbol{Y}^e} \left[ \ell_1(w(\text{Pa}(\boldsymbol{Y}^e)), \boldsymbol{Y}^e) \right], \quad (11)$$

where $\ell_1(\cdot)$ could be any loss. Since we assume that $\mathbb{E}(\boldsymbol{Y} | \text{Pa}(\boldsymbol{Y}))$ is invariant across $\mathcal{E}_{all}$ (the relaxed version of Assumption 1d), the learned $w$ is guaranteed to perform well across $\mathcal{E}_{all}$.

The remaining question is how to infer $\text{Pa}(\boldsymbol{Y})$ (i.e., $\boldsymbol{Z}_p$) from $\boldsymbol{X}$ in a new environment. This can be implemented by leveraging the learned $p(\boldsymbol{X} | \boldsymbol{Z})$. The rationale behind is that $p(\boldsymbol{X} | \boldsymbol{Z})$ is assumed to be invariant across $\mathcal{E}_{all}$ (Assumption 1c). In light of this idea, we follow Sun et al. (2020) and infer $\boldsymbol{Z}_p$ from $\boldsymbol{X}$ in any new testing environment by solving the following optimization problem:

$$\max_{\boldsymbol{Z}_p, \boldsymbol{Z}_c} \log p_{\boldsymbol{f}}(\boldsymbol{X} | \boldsymbol{Z}_p, \boldsymbol{Z}_c) + \lambda_1 ||\boldsymbol{Z}_p||_2^2 + \lambda_2 ||\boldsymbol{Z}_c||_2^2, \quad (12)$$

where the hyperparameters $\lambda_1 > 0$ and $\lambda_2 > 0$ control the learned $\boldsymbol{Z}_p$ and $\boldsymbol{Z}_c$ in a reasonable scale, and both are selected on training/validation data. For optimization, we follow Schott et al. (2018) to first use values of $\boldsymbol{Z}$ sampled from the training set as initial points and then use Adam to optimize for several iterations.[9] Note that the noise $\boldsymbol{\epsilon}$ will introduce uncertainty in the estimation of $\boldsymbol{Z}_p$ and $\boldsymbol{Z}_c$ from $\boldsymbol{X}$. However, as we mentioned before (below Theorem 3), this noise is not going to affect the estimation much because the likelihood will be highly concentrated around the ground truth values, as corroborated by our good empirical results.

A key question is if iCaRL performs well across $\mathcal{E}_{all}$ even though it uses only data from $\mathcal{E}_{tr}$. That is, does iCaRL enable OOD generalization, as defined by Arjovsky et al. (2019)? The answer is positive since Theorem A.1 in Arjovsky (2021) indicates that i) any predictor $w \circ \Phi$ with optimal OOD generalization uses only $\text{Pa}(\boldsymbol{Y})$ to compute $\Phi$ and ii) the classifier $w$ in this optimal predictor can be estimated using data from any environment $e$ for which the distribution of $\text{Pa}(\boldsymbol{Y}^e)$ has full support, which will always be the case since the conditional prior in Eq. (5) has full support. Finally, Theorem A.1 in Arjovsky (2021) also indicates that iii) the optimal predictor will be invariant across $\mathcal{E}_{all}$. Key to these results is that $\text{Pa}(\boldsymbol{Y}^e)$ are available when solving (12). This requires first to identify the latent variables $\boldsymbol{Z}$ from $\boldsymbol{X}$, $\boldsymbol{Y}$ and $\boldsymbol{E}$ and second, to discover the direct causes of $\boldsymbol{Y}$. The hypotheses of Theorems 1 and 2 and Assumption 1 provide this guarantee. We therefore have the following result whose proof is in Appendix H.

**Proposition 1.** *Under Assumption 1 and the assumptions of Theorems 1 and 2, the predictor learned by iCaRL across $\mathcal{E}_{tr}$ in the limit of infinite data has optimal OOD generalization across $\mathcal{E}_{all}$.*

## 5 Experiments

We compare our approach with a variety of methods on both synthetic and real-world datasets. In all comparisons, unless stated otherwise, we average performance over ten runs. **Due to space constraints, we only highlight some key results while pointing to the extensive appendices for more information.** The supplement contains all the details of the experiments, e.g., datasets (Appendix J), implementation (Appendix M), hyperparameters and architectures (Appendix N), etc.

### 5.1 Synthetic data

To verify the identifiability of NF-iVAE, we conduct a series of experiments on synthetic data generated according to the causal graph shown in Fig. 2e. Details of the ground truth data generating

---

[7]We only need to consider those variables in $\text{Ne}(\boldsymbol{Y})$ whose edges connecting to $\boldsymbol{Y}$ are not oriented by PC.

[8]These conditional independence tests can be performed in parallel to largely accelerate the testing procedure. See more in Appendix G.

[9]Note that, (12) can be optimized either independently for each data point or for a number of data points.

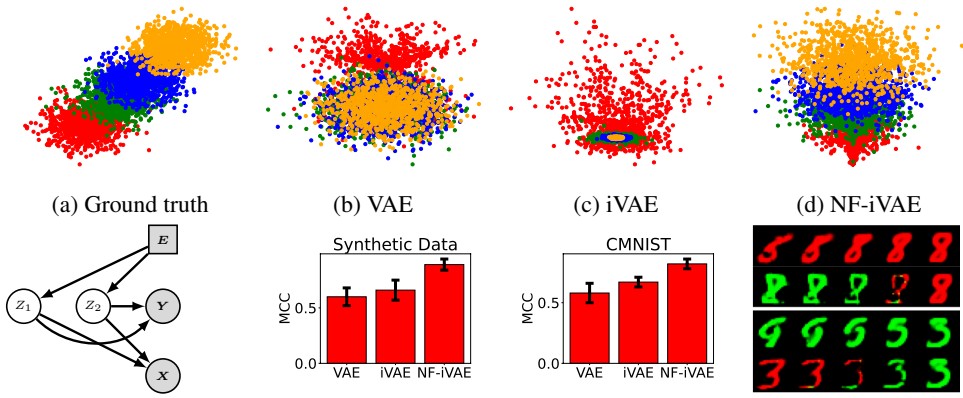

Figure 2: (a-d) Visualization of the samples (i.e., $\hat{\boldsymbol{Z}} = (\hat{Z}_1, \hat{Z}_2)$) in latent space recovered through different algorithms: (a) Samples from the true distribution; (b-c) Samples from the posterior inferred using VAE and iVAE, respectively. Apparently, our method (d) can recover the original data up to a permutation and a simple componentwise transformation. (e) The causal structure with $\boldsymbol{Y}$ having two causes describes the data generating process of the synthetic dataset. (f) Mean correlation coefficient (MCC) scores for VAE, iVAE, and NF-iVAE on synthetic data. (g) MCC scores for VAE, iVAE, and NF-iVAE on CMNIST. (h) The effects on the CMNIST images of digit 8 (top two rows) and digit 3 (bottom two rows) when intervening on a causal factor $Z_{i \in I_p}$ and on a non-causal factor (effect) $Z_{j \in I_c}$, respectively.

process are given in Appendix K. The reason we choose this setting is that it is the simplest case satisfying our requirements: a) For ease of visualization, the latent space had better be 2-dimensional; b) To introduce the non-factorized prior given $\boldsymbol{Y}$ and $\boldsymbol{E}$ (i.e., $Z_i \not\perp\!\!\!\perp Z_j | \boldsymbol{Y}, \boldsymbol{E}$), $\boldsymbol{Y}$ has at least two causes. We draw 1000 samples from each of the four environments $\boldsymbol{E} = \{0.2, 2, 3, 5\}$, and thus the whole synthetic dataset consists of 4000 samples. The task is to recover the true latent variable $\boldsymbol{Z} = (Z_1, Z_2)$ using the samples of $\boldsymbol{X}$, $\boldsymbol{E}$, and $\boldsymbol{Y}$. We compare with two widely-used baselines: VAE (Kingma & Welling, 2013) (without identifiability guarantees) and iVAE (Khemakhem et al., 2020a) (with a conditionally factorized prior for identifiability). Through the aforementioned theoretical analysis, it is evident that our method has a more general assumption on the prior leading to identifiability. This is demonstrated empirically in Figs. 2b-2d. Our method NF-iVAE can recover the original data $\boldsymbol{Z}$ up to a permutation and a simple componentwise transformation, whereas all the other methods fail because they are unable to handle the non-factorized case in which $Z_i \not\perp\!\!\!\perp Z_j | \boldsymbol{Y}, \boldsymbol{E}$. We also compute the mean correlation coefficient (MCC) used in Khemakhem et al. (2020a), which can be obtained by calculating the correlation coefficient between all pairs of true and recovered latent factors and then solving a linear sum assignment problem by assigning each recovered latent factor to the true latent factor with which it best correlates. By definition, higher MCC scores indicate stronger identifiability. From Fig. 2f, we can see that the MCC score for NF-iVAE is significantly greater than those of VAE and iVAE, indicating much stronger identifiability. Note that, in Appendix K we additionally compare with more methods, whose differences are further summarized in a table.

## 5.2 COLORED MNIST, COLORED FASHION MNIST, AND VLCS

In this section, we first report experiments on two datasets used in IRM and IRMG: Colored MNIST (CMNIST) and Colored Fashion MNIST (CFMNIST). We follow the same setting of Ahuja et al. (2020a) to create these two datasets (see the details in Appendix J). The task is to predict a binary label assigned to each image which is originally grayscale but artificially colored in a way that the color is correlated strongly but spuriously with the class label. For all the experiments on these two datasets, we set the number of the latent variables to $n = 10$.

Likewise, we investigate the identifiability of NF-iVAE on CMNIST by computing the MCC score between samples of the true latent variable and of the recovered latent variable. Since the true latent variable on CMNIST is inaccessible to us, we follow Khemakhem et al. (2020b) and compute an average MCC score between samples of latent variables recovered by different models trained with different random initialization. As shown in Fig. 2g, it is evident that the MCC score for NF-iVAE greatly outperforms the others, showing that the latent variables recovered by NF-iVAE have much better identifiability.

Furthermore, we demonstrate the ability of iCaRL to discover the causal latent variables (Phase 2) by visualizing the generated images through performing intervention upon a causal latent variable and a non-causal latent variable, respectively. Fig. 2h shows how intervening upon each of them affects the image. Obviously, intervening on a causal latent variable affects the shape of the digit but not its color

Table 1: Colored Fashion MNIST. Comparisons in terms of accuracy (%) (mean ± std deviation).

| METHOD | TRAIN | TEST |
|---|---|---|
| ERM | $83.17 \pm 1.01$ | $22.46 \pm 0.68$ |
| ERM 1 | $81.33 \pm 1.35$ | $33.34 \pm 8.85$ |
| ERM 2 | $84.39 \pm 1.89$ | $13.16 \pm 0.82$ |
| ROBUST MIN MAX | $82.81 \pm 0.11$ | $29.22 \pm 8.56$ |
| F-IRM GAME | $62.31 \pm 2.35$ | $69.25 \pm 5.82$ |
| V-IRM GAME | $68.96 \pm 0.95$ | $70.19 \pm 1.47$ |
| IRM | $75.01 \pm 0.25$ | $55.25 \pm 12.42$ |
| **iCaRL (ours)** | $\mathbf{74.96 \pm 0.37}$ | $\mathbf{73.61 \pm 0.63}$ |
| ERM GRAYSCALE | $74.79 \pm 0.37$ | $74.67 \pm 0.48$ |
| OPTIMAL | $75$ | $75$ |

Table 2: VLCS. Comparisons in terms of accuracy (%) (mean ± std deviation).

| METHOD | TEST |
|---|---|
| ERM | $77.4 \pm 0.3$ |
| IRM | $78.1 \pm 0.0$ |
| DRO (Sagawa et al., 2019) | $77.2 \pm 0.6$ |
| Mixup (Yan et al., 2020) | $77.7 \pm 0.4$ |
| CORAL (Sun & Saenko, 2016) | $77.7 \pm 0.5$ |
| MMD (Li et al., 2018b) | $76.7 \pm 0.9$ |
| DANN (Ganin et al., 2016) | $78.7 \pm 0.3$ |
| C-DANN (Li et al., 2018c) | $78.2 \pm 0.4$ |
| LaCIM (Sun et al., 2020) | $78.4 \pm 0.5$ |
| **iCaRL (ours)** | $\mathbf{81.8 \pm 0.6}$ |

(top plots), whilst intervening on a non-causal latent variable, which is an effect in Fig. 2h, affects the color of the digit only (bottom plots). This visually verifies the results of iCaRL in Phase 2.

In terms of the OOD generalization performance, we compare iCaRL with 1) IRM, 2) two variants of IRMG: F-IRM Game (with $\Phi$ fixed to the identity) and V-IRM Game (with a variable $\Phi$), 3) three variants of ERM: ERM (on entire training data), ERM $e$ (on each environment $e$), and ERM GRAYSCALE (on data with no spurious correlations), and 4) ROBUST MIN MAZ (minimizing the maximum loss across the multiple environments). Table 1 shows that iCaRL outperforms all other baselines on CFMNIST. It is worth emphasising that the train and test accuracies of iCaRL closely approach the ones of ERM GRAYSCALE and OPTIMAL, implying that iCaRL approximately learns the true invariant causal representations with almost no correlation with the spurious color feature. We can draw similar conclusions from the results on CMNIST (Appendix L).

We also report the results on one of the widely used realistic datasets for OOD generalization: VLCS (Fang et al., 2013). This dataset consists of $10,729$ photographic images of dimension $(3, 224, 224)$ and $5$ classes from four domains: Caltech101, LabelMe, SUN09, and VOC2007. We used the exact experimental setting that is described in Gulrajani & Lopez-Paz (2020). We provide results averaged over all possible train and test environment combination for one of the commonly used hyper-parameter tuning procedure: train domain validation. As shown in Table 2, iCaRL achieves state-of-the-art performance when compared to those most popular domain generalization alternatives. We further include experimental evidence on another popular dataset: PACS (Li et al., 2017a), all the details of which are placed in Appendix L.

## 6 RELATED WORK

Invariant Causal Prediction (ICP), aims to find the *causal feature set* (i.e., all direct causes of a target variable of interest) (Peters et al., 2015) by exploiting the invariance property in causality which has been discussed under the term "autonomy", "modularity", and "stability" (Haavelmo, 1944; Aldrich, 1989; Hoover, 1990; Pearl, 2009; Dawid et al., 2010; Schölkopf et al., 2012). This invariance property assumed in ICP and its nonlinear extension (Heinze-Deml et al., 2018) is limited, because no intervention is allowed on the target variable $Y$. Besides, ICP methods implicitly assume that the variables of interest $Z$ are given. The works of Magliacane et al. (2018) and Subbaswamy et al. (2019) attempt to find invariant predictors that are maximally predictive using conditional independence tests and other graph-theoretic tools, both of which also assume that the $Z$ are given and further assume that additional information about the structure over $Z$ is known. Mitrovic et al. (2020) analyze data augmentations in self-supervised learning from the perspective of invariant causal mechanisms. Arjovsky et al. (2019) reformulate this invariance as an optimization-based problem, allowing us to learn an invariant data representation from $X$ constrained to be a linear transformation of $Z$. The risks of this approach have been discussed in Rosenfeld et al. (2020); Kamath et al. (2021); Nagarajan et al. (2020) and its sample complexity is analyzed in Ahuja et al. (2020b).

Another line of related work is in the field of domain generalization, which we discuss in Appendix A.

## 7 CONCLUSION

We have proposed a novel framework to learn invariant predictors from a diverse set of training environments. It is based on a practical and general assumption: the prior over the data representation belongs to a general exponential family when conditioning on the target and the environment. This assumption leads to guarantees that the components in the representation can be identified up to a permutation and simple transformation. This allows us to discover all the direct causes of the target, which enables generalization guarantees in the nonlinear setting. We hope our framework will inspire new ways to address the OOD generalization problem through a causal lens.

ACKNOWLEDGEMENTS

We are thankful to Wenlin Chen for contributing to Step III in the proof of Theorem 4 and for generalizing the proof in Theorem 5 to the case in which each $T_i(Z_i)$ in $T_f(Z)$ contains arbitrary sufficient statistics instead of just $Z_i$ and $Z_i^2$. We also thank Ilyes Khemakhem, Aapo Hyvärinen, and Arthur Gretton for their helpful discussions, and the anonymous reviewers for their constructive comments on an earlier version of this paper.

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

# A   DOMAIN GENERALIZATION

The goal of domain generalization (DG) (Muandet et al., 2013) is OOD generalization: learning a predictor that performs well at unseen test domains. Unlike domain adaptation (Pan & Yang, 2009; Ben-David et al., 2007; 2010; Crammer et al., 2008; Patel et al., 2015; Zhao et al., 2019; Wilson & Cook, 2020; Zhang et al., 2015), DG assumes that the test domain data are not available during training. One thread of DG is to explore techniques from kernel methods (Muandet et al., 2013; Niu et al., 2015; Erfani et al., 2016; Li et al., 2017b). Muandet et al. (2013) propose a kernel-based optimization algorithm that learns an invariant transformation by minimizing the discrepancy among domains and preventing the loss of relationship between input and output features. Another line of DG work is using end-to-end methods from deep learning: (a) reducing the differences of representations across domains through adversarial or similar techniques (Ghifary et al., 2015; Wang et al., 2017; Motiian et al., 2017; Li et al., 2018b;c); (b) projecting out superficial domain-specific statistics to reduce sensitivity to the domain (Wang et al., 2019); (c) fusing representations from an ensemble of models across domains (Ding & Fu, 2017; Mancini et al., 2018). Meta-learning can also be applied to domain generalization, by dividing source domains into meta-training and meta-test sets, and aiming for a low generalization error on meta-test sets after training on meta-training sets (Balaji et al., 2018; Dou et al., 2019; Li et al., 2018a; 2019a;b). Recently, an extensive empirical survey of many DG algorithm (Gulrajani & Lopez-Paz, 2020) suggested that with current models and data augmentation techniques, plain ERM may be competitive with the state-of-the-art. It is worth noting that Sun et al. (2020) also propose an approach to learning latent causal factors for prediction. However, their assumptions over the underlying causal graph are restricted due to two reasons: 1) they only consider the scenarios when $Z$ and $Y$ are generated concurrently, which excludes the cases in which some part of $Z$ could also be affected by $Y$ in some manner; 2) They assume that the causal latent factors $Z_p$ and the non-causal latent factors $Z_c$ are independent when conditioning on $E$, that is, $Z_p \perp\!\!\!\perp Z_c | E$. In practice, during model learning, they actually further assume that $Z_i \perp\!\!\!\perp Z_j | E$ for any $i \neq j$ so that VAE could be leveraged to learn the model. In this sense, their approach has the same issue as the one in iVAE, i.e., unable to deal with the non-factorized cases. This point is also verified in Table 2, where our approach greatly outperforms theirs.

# B   VARIATIONAL AUTOENCODERS

We briefly describe the framework of variational autoencoders (VAEs), which allows us to efficiently learn deep latent-variable models and their corresponding inference models (Kingma & Welling, 2013; Rezende et al., 2014). Consider a simple latent variable model where $X \in \mathbb{R}^d$ stands for an observed variable and $Z \in \mathbb{R}^n$ for a latent variable. A VAE method learns a full generative model $p_{\boldsymbol{\theta}}(X, Z) = p_{\boldsymbol{\theta}}(X|Z)p_{\boldsymbol{\theta}}(Z)$ and an inference model $q_{\boldsymbol{\phi}}(Z|X)$, typically a factorized Gaussian distribution whose mean and variance parameters are given by the output of a neural network with input $X$. This inference model approximates the posterior $p_{\boldsymbol{\theta}}(Z|X)$, where $\boldsymbol{\theta}$ is a vector of parameters of the generative model, $\boldsymbol{\phi}$ a vector of parameters of the inference model, and $p_{\boldsymbol{\theta}}(Z)$ is a prior distribution over the latent variables. Instead of maximizing the data log-likelihood, we maximize its lower bound $\mathcal{L}_{\mathrm{VAE}}(\boldsymbol{\theta}, \boldsymbol{\phi})$:

$$\log p_{\boldsymbol{\theta}}(X) \geq \mathcal{L}_{\mathrm{VAE}}(\boldsymbol{\theta}, \boldsymbol{\phi}) := \mathbb{E}_{q_{\boldsymbol{\phi}}(Z|X)}\left[\log p_{\boldsymbol{\theta}}(X|Z)\right] - \mathrm{KL}\left(q_{\boldsymbol{\phi}}(Z|X)||p_{\boldsymbol{\theta}}(Z)\right),$$

where we have used Jensen's inequality, and $\mathrm{KL}(\cdot||\cdot)$ denotes the Kullback-Leibler divergence between two distributions.

# C   DERIVATION

## C.1   iVAE

In Section 2.1, the evidence lower bound of iVAE is defined by

$$\begin{aligned}\mathcal{L}_{\mathrm{iVAE}}(\boldsymbol{\theta}, \boldsymbol{\phi}) :=& \mathbb{E}_{p_D}\left[\mathbb{E}_{q_{\boldsymbol{\phi}}(Z|X,U)}\left[\log p_{\boldsymbol{f}}(X|Z)\right] - \mathrm{KL}\left(q_{\boldsymbol{\phi}}(Z|X,U)||p_{\boldsymbol{T},\boldsymbol{\lambda}}(Z|U)\right)\right] \\ =& \mathbb{E}_{p_D}\left[\mathbb{E}_{q_{\boldsymbol{\phi}}(Z|X,U)}\left[\log p_{\boldsymbol{f}}(X|Z) + \log p_{\boldsymbol{T},\boldsymbol{\lambda}}(Z|U) - \log q_{\boldsymbol{\phi}}(Z|X,U)\right]\right].\end{aligned}$$

## C.2 Score Matching

In Section 4.1, we follow Hyvärinen (2005) and use a simple trick of partial integration to simplify the evaluation of the score matching objective $\mathcal{L}_{\text{phase1}}^{\text{SM}}$ in Eq. (10) of the main text:

$$
\begin{aligned}
\mathcal{L}_{\text{phase1}}^{\text{SM}}(\boldsymbol{T}, \boldsymbol{\lambda}) :=& \mathbb{E}_{p_D} \left[ \mathbb{E}_{q_{\boldsymbol{\phi}}(\boldsymbol{Z}|\boldsymbol{X},\boldsymbol{Y},\boldsymbol{E})} \left[ ||\nabla_{\boldsymbol{Z}} \log q_{\boldsymbol{\phi}}(\boldsymbol{Z}|\boldsymbol{X},\boldsymbol{Y},\boldsymbol{E}) - \nabla_{\boldsymbol{Z}} \log p_{\boldsymbol{T},\boldsymbol{\lambda}}(\boldsymbol{Z}|\boldsymbol{Y},\boldsymbol{E})||^2 \right] \right] \\
=& \mathbb{E}_{p_D} \left[ \mathbb{E}_{q_{\boldsymbol{\phi}}(\boldsymbol{Z}|\boldsymbol{X},\boldsymbol{Y},\boldsymbol{E})} \left[ \sum_{j=1}^{n} \left[ \frac{\partial^2 p_{\boldsymbol{T},\boldsymbol{\lambda}}(\boldsymbol{Z}|\boldsymbol{Y},\boldsymbol{E})}{\partial Z_j^2} + \frac{1}{2} \left( \frac{\partial p_{\boldsymbol{T},\boldsymbol{\lambda}}(\boldsymbol{Z}|\boldsymbol{Y},\boldsymbol{E})}{\partial Z_j} \right)^2 \right] \right] \right] + const.
\end{aligned}
$$

where the last equality is due to the Theorem 1 in Hyvärinen (2005).

## D   Definitions for SEM and IRM

**Definition 2.** *A structural equation model (SEM) $\mathcal{M} := (\mathcal{S}, N)$ governing the random vector $\boldsymbol{Z} = (Z_1, \ldots, Z_d)$ is a set of structural equations:*

$$\mathcal{S}_i : Z_i \leftarrow f_i(\text{Pa}(Z_i), N_i),$$

*where $\text{Pa}(Z_i) \subseteq \{Z_1, \ldots, Z_d\} \setminus \{Z_i\}$ are called the parents of $Z_i$, and the $N_i$ are independent noise random variables. We say that "$Z_i$ causes $Z_j$" if $Z_i \in \text{Pa}(Z_j)$. We call causal graph of $\boldsymbol{Z}$ to the graph obtained by drawing i) one node for each $Z_i$, and ii) one edge from $Z_i$ to $Z_j$ if $Z_i \in \text{Pa}(Z_j)$. We assume acyclic causal graphs.*

**Definition 3.** *Consider a SEM $\mathcal{M} := (\mathcal{S}, N)$. An intervention $e$ on $\mathcal{M}$ consists of replacing one or several of its structural equations to obtain an intervened SEM $\mathcal{M}^e := (\mathcal{S}^e, N^e)$, with structural equations:*

$$\mathcal{S}_i^e : Z_i^e \leftarrow f_i^e(\text{Pa}^e(Z_i^e), N_i^e),$$

*The variable $\boldsymbol{Z}^e$ is intervened if $\mathcal{S}_i \neq \mathcal{S}_i^e$ or $N_i \neq N_i^e$.*

**Definition 4.** *Consider a structural equation model (SEM) $\mathcal{S}$ governing the random vector $(Z_1, \ldots, Z_n, \boldsymbol{Y})$, and the learning goal of predicting $\boldsymbol{Y}$ from $\boldsymbol{Z}$. Then, the set of all environments $\mathcal{E}_{all}(\mathcal{S})$ indexes all the interventional distributions $P(\boldsymbol{Z}^e, \boldsymbol{Y}^e)$ obtainable by valid interventions $e$. An intervention $e \in \mathcal{E}_{all}(\mathcal{S})$ is valid as long as (i) the causal graph remains acyclic, (ii) $\mathbb{E}[\boldsymbol{Y}^e|\text{Pa}(\boldsymbol{Y})] = \mathbb{E}[\boldsymbol{Y}|\text{Pa}(\boldsymbol{Y})]$, and (iii) $\mathbb{V}[\boldsymbol{Y}^e|\text{Pa}(\boldsymbol{Y})]$ remains within a finite range.*

## E   Definitions and Lemmas for the Exponential Families

**Definition 5.** *(Exponential family) A multivariate exponential family is a set of distributions whose probability density function can be written as*

$$p(\boldsymbol{Z}) = \frac{\mathcal{Q}(\boldsymbol{Z})}{\mathcal{C}(\boldsymbol{\theta})} \exp(\langle \boldsymbol{T}(\boldsymbol{Z}), \boldsymbol{\theta} \rangle), \tag{13}$$

*where $\mathcal{Q} : \mathcal{Z} \to \mathbb{R}$ is the base measure, $\mathcal{C}(\boldsymbol{\theta})$ is the normalizing constant, $\boldsymbol{T} : \mathcal{Z} \to \mathbb{R}^k$ is the sufficient statistics, and $\boldsymbol{\theta} \in \mathbb{R}^k$ is the natural parameter. The size $k \geq n$ is the dimension of the sufficient statistics $\boldsymbol{T}$ and depends on the latent space dimension $n$. Note that $k$ is fixed given $n$.*

**Definition 6.** *(Strongly exponential distributions) A multivariate exponential family distribution*

$$p(\boldsymbol{Z}) = \frac{\mathcal{Q}(\boldsymbol{Z})}{\mathcal{C}(\boldsymbol{\theta})} \exp(\langle \boldsymbol{T}(\boldsymbol{Z}), \boldsymbol{\theta} \rangle) \tag{14}$$

*is strongly exponential, if*

$$(\exists \boldsymbol{\theta} \in \mathbb{R}^k \ s.t. \ \langle \boldsymbol{T}(\boldsymbol{Z}), \boldsymbol{\theta} \rangle = const, \ \forall \boldsymbol{Z} \in \mathcal{Z}) \implies (l(\mathcal{Z}) = 0 \ or \ \boldsymbol{\theta} = \boldsymbol{0}), \quad \forall \mathcal{Z} \subset \mathbb{R}^n, \tag{15}$$

*where $l$ is the Lebesgue measure.*

The density of a strongly exponential distribution has almost surely the exponential component and can only be reduced to the base measure on a set of measure zero. Note that all common multivariate exponential family distributions (e.g. multivariate Gaussian) are strongly exponential.

## F   Definitions for Identifiability

**Definition 7.** *Let $\Theta$ be the domain of the parameters $\boldsymbol{\theta} = \{\boldsymbol{f}, \boldsymbol{T}, \boldsymbol{\lambda}\}$. Let $\sim$ be an equivalence relation on $\Theta$. A deep generative model is said to be $\sim$–identifiable if*

$$p_{\boldsymbol{\theta}}(\boldsymbol{X}) = p_{\tilde{\boldsymbol{\theta}}}(\boldsymbol{X}) \implies \boldsymbol{\theta} \sim \tilde{\boldsymbol{\theta}}. \tag{16}$$

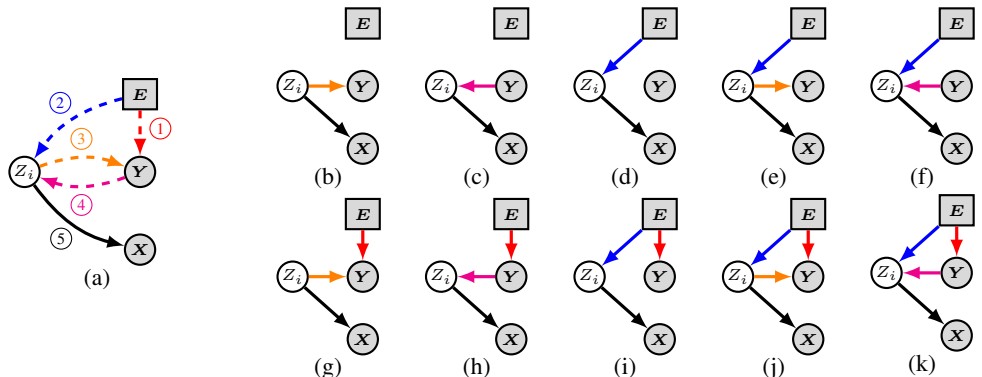

Figure 3: (a) General causal structure over $\{X_i, \boldsymbol{Y}, \boldsymbol{X}, \boldsymbol{E}\}$, where the arrow from $Z_i$ to $\boldsymbol{X}$ is a must-have connection and the other four might not be necessary. (b) Ten possible causal structures from (a) under Assumptions 1&2.

*The elements in the quotient space $\Theta \backslash \sim$ are called the identifiability classes.*

**Definition 8.** *Let $\sim_A$ be an equivalence relation on $\Theta$ defined by:*

$$(\boldsymbol{f}, \boldsymbol{T}, \boldsymbol{\lambda}) \sim_A (\tilde{\boldsymbol{f}}, \tilde{\boldsymbol{T}}, \tilde{\boldsymbol{\lambda}}) \iff \exists A, \boldsymbol{c} \text{ s.t. } \boldsymbol{T}(\boldsymbol{f}^{-1}(\boldsymbol{X})) = A\tilde{\boldsymbol{T}}(\tilde{\boldsymbol{f}}^{-1}(\boldsymbol{X})) + \boldsymbol{c}, \ \forall \boldsymbol{X} \in \mathcal{O}, \quad (17)$$

*where $A \in \mathbb{R}^{k \times k}$ is an invertible matrix and $\boldsymbol{c} \in \mathbb{R}^k$ is a vector.*

**Definition 9.** *Let $\sim_P$ be an equivalence relation on $\Theta$ defined by:*

$$(\boldsymbol{f}, \boldsymbol{T}, \boldsymbol{\lambda}) \sim_P (\tilde{\boldsymbol{f}}, \tilde{\boldsymbol{T}}, \tilde{\boldsymbol{\lambda}}) \iff \exists P, \boldsymbol{c} \text{ s.t. } \boldsymbol{T}(\boldsymbol{f}^{-1}(\boldsymbol{X})) = P\tilde{\boldsymbol{T}}(\tilde{\boldsymbol{f}}^{-1}(\boldsymbol{X})) + \boldsymbol{c}, \ \forall \boldsymbol{X} \in \mathcal{O}, \quad (18)$$

*where $P \in \mathbb{R}^{k \times k}$ is a block permutation matrix and $\boldsymbol{c} \in \mathbb{R}^k$ is a vector.*

## G  PHASE 2: DISCOVERING SINGLE CAUSE

### G.1  SOME SPECIAL CASES IN MULTI-CAUSE SETTINGS

These conditional independence tests can be performed in parallel to largely accelerate the testing procedure. Note that, in practice, it might occur that there exist some $Z_i$ which is independent of any other $Z_j$ when conditioning on $\boldsymbol{Y}$ and $\boldsymbol{E}$. It is probably because such $Z_i$ is a deterministic transformation of $\boldsymbol{Y}$. In this special case, we can use IGCI (Daniusis et al., 2012; Janzing et al., 2012) to determine whether or not $Z_i$ is a cause of $\boldsymbol{Y}$. Also note that, in some scenarios in which the dependence signals between a pair of causal latent variables might be weak due to the data issue, we can test the conditional independence of such a latent variable with all the other causal latent variables. If it is conditionally dependent on more than half of them or its average p-value is larger than the pre-specified threshold, we will select it as one cause of $\boldsymbol{Y}$.

### G.2  DISCOVERING SINGLE CAUSE

In the single-cause case, by following Wang et al. (2014), we leverage the *MB-by-MB* algorithm to first construct a *local* structure around $\boldsymbol{Y}$ and then discover the single parent of $\boldsymbol{Y}$ according to the constructed *local* graph. One obvious advantage of this approach is in efficiency, because there is no need to construct the whole causal graph containing all the latent variables and $\boldsymbol{Y}$.

We could have an even more efficient solution to the single-cause case in some special scenarios where we assume that $Z_i \perp\!\!\!\perp Z_j | \boldsymbol{Y}, \boldsymbol{E}$ for any $i \neq j$. In fact, this assumption covers more scenarios than the common assumption that $Z_i \perp\!\!\!\perp Z_j$ for $i \neq j$ in latent variable models, e.g., disentanglement (Bengio et al., 2013; Locatello et al., 2019), autoencoders (Kingma & Welling, 2013; Rezende et al., 2014), ICA (Comon, 1994; Hyvärinen & Oja, 2000), etc. If $\boldsymbol{Y}$ is caused by at most one of $Z_i$ and $Z_j$, and no matter whether $Z_i$ and $Z_j$ are caused by $\boldsymbol{E}$ or not, then $Z_i \perp\!\!\!\perp Z_j | \boldsymbol{Y}, \boldsymbol{E}$ holds, but we may well have $Z_i \not\!\perp\!\!\!\perp Z_j$ (e.g., if $\boldsymbol{Y}$ causes both $Z_i$ and $Z_j$, or if there is a chain $Z_i \rightarrow \boldsymbol{Y} \rightarrow Z_j$). If $\boldsymbol{Y}$ causes or is caused by at most one of $Z_i$ and $Z_j$, and at most one of $Z_i$ and $Z_j$ is caused by $\boldsymbol{E}$, then both $Z_i \perp\!\!\!\perp Z_j$ and $Z_i \perp\!\!\!\perp Z_j | \boldsymbol{Y}, \boldsymbol{E}$ hold. If $\{\boldsymbol{Y}, Z_i, Z_j\}$ form a collider $Z_i \rightarrow \boldsymbol{Y} \leftarrow Z_j$, and no matter whether $Z_i$ and $Z_j$ are caused by $\boldsymbol{E}$ or not, then $Z_i \perp\!\!\!\perp Z_j | \boldsymbol{E}$ hold, but we may have $Z_i \not\!\perp\!\!\!\perp Z_j$ (e.g., when both $Z_i$ and $Z_j$ are caused by $\boldsymbol{E}$).

Under this assumption, we are able to separately look into each $Z_i$ given $\boldsymbol{Y}$ and $\boldsymbol{E}$, without considering any other $Z_j$. Fig. 5a shows that there exist only five possible connections between $Z_i$, $\boldsymbol{Y}$, $\boldsymbol{E}$, and $\boldsymbol{X}$. Among them, only the arrow from $Z_i$ to $\boldsymbol{X}$ must exist, because $\boldsymbol{X}$ is generated from $\boldsymbol{Z}$. The other four arrows might not be present, with the exception that there must be at least one connection between $Z_i$ and $\boldsymbol{Y}$ or $\boldsymbol{E}$ (Assumption 1a). This leaves ten possible structures, shown in Figs. 3b-3k.

Given data $\{\hat{Z}_i, \boldsymbol{Y}, \boldsymbol{E}, \boldsymbol{X}\}$ in which the $\hat{Z}_i$ are obtained using $q_\phi(\boldsymbol{Z}|\boldsymbol{X}, \boldsymbol{Y}, \boldsymbol{E})$ (for example, as given by the mean of this distribution), we are able to distinguish all ten structures in Figs. 3b-3k by using causal discovery algorithms (Peters et al., 2017; Zhang et al., 2017; Huang et al., 2020) and performing conditional independence tests (Spirtes et al., 2000; Zhang et al., 2012). This is summarized in Proposition 2 below. Its proof can be found in Appendix H, which also describes the specific assumptions made. In practice, we can assess in parallel whether or not each $Z_i$ is a direct cause of $\boldsymbol{Y}$, which accelerates this phase significantly.

**Proposition 2.** *Under the assumptions stated in Appendix H, the ten structures shown in Figs. 3b-3k can be identified using causal discovery methods consistent in the infinite sample limit.*

Note that there are only four cases in which $Z_i$ is a parent of $\boldsymbol{Y}$ (i.e., Figs. 3b, 3e, 3g, and 3j). We can identify these by applying rules **1.2**, **1.6**, **2.1**, and **3.1** from Appendix H.5.

## H  PROOFS

### H.1  PROOF OF THEOREM 1

The proof of this theorem consists of three parts.

In Part I, we prove that the parameters $\boldsymbol{\theta}$ are $\sim_A$ identifiable (Definition 8) by using assumption (i), the first half of assumption (ii), and assumption (iv) of Theorem 1.

In Part II, based on the result in Part I, we further prove that the parameters $\boldsymbol{\theta}$ are $\sim_P$ identifiable (Definition 9) by additionally using Assumption 2, the second half of assumption (ii) and assumption (iii) of Theorem 1.

In Part III, we combine the results (Theorems 4&5) in both Part I and Part II into one theorem (Theorem 1), which completes the proof.

It is worth noting that, compared to the proof in iVAE, the main changes in our proof consist of

- Part I, In step III.
  - It has been updated to account for vectors of sufficient statistics whose entries can be arbitrary functions of all entries in the random variable vector, while in the previous proof the sufficient statistics contained entries that are functions of individual entries in the random variable vector.
  - The assumption of "The sufficient statistics in $\boldsymbol{T}$ are all linearly independent." is not required in our proof, but it is in the proof of iVAE.
- Part II.
  - It has been updated to account for the part of the sufficient statistics which is the output of a deep neural network with ReLU activation functions.

### H.1.1  PART I

For notational simplicity, in the proof of this part we denote $(\boldsymbol{Y}, \boldsymbol{E})$ by $\boldsymbol{U}$. Hence, our generative model defined according to Eqs. (6-8) in the main text now becomes:

$$p_{\boldsymbol{\theta}}(\boldsymbol{X}, \boldsymbol{Z}|\boldsymbol{U}) = p_{\boldsymbol{f}}(\boldsymbol{X}|\boldsymbol{Z})p_{\boldsymbol{T},\boldsymbol{\lambda}}(\boldsymbol{Z}|\boldsymbol{U}), \tag{19}$$

$$p_{\boldsymbol{f}}(\boldsymbol{X}|\boldsymbol{Z}) = p_{\boldsymbol{\epsilon}}(\boldsymbol{X} - \boldsymbol{f}(\boldsymbol{Z})), \tag{20}$$

$$p_{\boldsymbol{T},\boldsymbol{\lambda}}(\boldsymbol{Z}|\boldsymbol{U}) = \mathcal{Q}(\boldsymbol{Z})/\mathcal{C}(\boldsymbol{U}) \exp\left[\boldsymbol{T}(\boldsymbol{Z})^{\mathrm{T}}\boldsymbol{\lambda}(\boldsymbol{U})\right]. \tag{21}$$

**Theorem 4.** *Suppose that we observe data sampled from a deep generative model defined according to Eqs. (19-21) with parameters $(\boldsymbol{f}, \boldsymbol{T}, \boldsymbol{\lambda})$. Assume that*

(i) *The set $\{\boldsymbol{X} \in \mathbb{O}|\varphi_{\boldsymbol{\varepsilon}}(\boldsymbol{X}) = 0\}$ has measure zero, where $\varphi_{\boldsymbol{\varepsilon}}$ is the characteristic function of the density $p_{\boldsymbol{\varepsilon}}$ defined in Eq. (20);*

*(ii)  The mixing function $\boldsymbol{f}$ in Eq. (20) is injective;*

*(iii)  There exist $k+1$ points $\boldsymbol{U}^0, \boldsymbol{U}^1, \cdots, \boldsymbol{U}^k \in \mathcal{U}$ such that the matrix*
$$L = [\boldsymbol{\lambda}(\boldsymbol{U}^1) - \boldsymbol{\lambda}(\boldsymbol{U}^0), \cdots, \boldsymbol{\lambda}(\boldsymbol{U}^k) - \boldsymbol{\lambda}(\boldsymbol{U}^0)] \in \mathbb{R}^{k \times k} \qquad (22)$$
*is invertible.*

*Then the parameters $\{\boldsymbol{f}, \boldsymbol{T}, \boldsymbol{\lambda}\}$ are $\sim_A$ identifiable.*

*Proof.* Define $\mathrm{vol}(B) = \sqrt{\det(B^T B)}$ for any full column rank matrix $B$. Suppose that we have two sets of parameters $\boldsymbol{\theta} = (\boldsymbol{f}, \boldsymbol{T}, \boldsymbol{\lambda})$ and $\tilde{\boldsymbol{\theta}} = (\tilde{\boldsymbol{f}}, \tilde{\boldsymbol{T}}, \tilde{\boldsymbol{\lambda}})$ such that $p_{\boldsymbol{\theta}}(\boldsymbol{X}|\boldsymbol{U}) = p_{\tilde{\boldsymbol{\theta}}}(\boldsymbol{X}|\boldsymbol{U})$, $\forall (\boldsymbol{X}, \boldsymbol{U}) \in \mathcal{O} \times \mathcal{U}$. We want to show $\boldsymbol{\theta} \sim_A \tilde{\boldsymbol{\theta}}$.

**Step I.** The proof of this step is similar to Step I in the proof of Theorem 1 in Khemakhem et al. (2020a). We transform the equality of the marginal distributions over observed data into the equality of noise-free distributions. For all pairs $(\boldsymbol{X}, \boldsymbol{U}) \in \mathcal{O} \times \mathcal{U}$, we have
$$p_{\boldsymbol{\theta}}(\boldsymbol{X}|\boldsymbol{U}) = p_{\tilde{\boldsymbol{\theta}}}(\boldsymbol{X}|\boldsymbol{U}) \qquad (23)$$
$$\implies \int_{\mathcal{Z}} p_{\boldsymbol{f}}(\boldsymbol{X}|\boldsymbol{Z}) p_{\boldsymbol{T},\boldsymbol{\lambda}}(\boldsymbol{Z}|\boldsymbol{U}) d\boldsymbol{Z} = \int_{\mathcal{Z}} p_{\tilde{\boldsymbol{f}}}(\boldsymbol{X}|\boldsymbol{Z}) p_{\tilde{\boldsymbol{T}},\tilde{\boldsymbol{\lambda}}}(\boldsymbol{Z}|\boldsymbol{U}) d\boldsymbol{Z} \qquad (24)$$
$$\implies \int_{\mathcal{Z}} p_{\boldsymbol{\varepsilon}}(\boldsymbol{X} - \boldsymbol{f}(\boldsymbol{Z})) p_{\boldsymbol{T},\boldsymbol{\lambda}}(\boldsymbol{Z}|\boldsymbol{U}) d\boldsymbol{Z} = \int_{\mathcal{Z}} p_{\boldsymbol{\varepsilon}}(\boldsymbol{X} - \tilde{\boldsymbol{f}}(\boldsymbol{Z})) p_{\tilde{\boldsymbol{T}},\tilde{\boldsymbol{\lambda}}}(\boldsymbol{Z}|\boldsymbol{U}) d\boldsymbol{Z} \qquad (25)$$
$$\implies \int_{\mathcal{O}} p_{\boldsymbol{\varepsilon}}(\boldsymbol{X} - \bar{\boldsymbol{X}}) p_{\boldsymbol{T},\boldsymbol{\lambda}}(\boldsymbol{f}^{-1}(\bar{\boldsymbol{X}})|\boldsymbol{U}) \mathrm{vol}(J_{\boldsymbol{f}^{-1}}(\bar{\boldsymbol{X}})) d\bar{\boldsymbol{X}} = \int_{\mathcal{O}} p_{\boldsymbol{\varepsilon}}(\boldsymbol{X} - \bar{\boldsymbol{X}}) p_{\tilde{\boldsymbol{T}},\tilde{\boldsymbol{\lambda}}}(\tilde{\boldsymbol{f}}^{-1}(\bar{\boldsymbol{X}})|\boldsymbol{U}) \mathrm{vol}(J_{\tilde{\boldsymbol{f}}^{-1}}(\bar{\boldsymbol{X}})) d\bar{\boldsymbol{X}} \qquad (26)$$
$$\implies \int_{\mathbb{R}^d} p_{\boldsymbol{\varepsilon}}(\boldsymbol{X} - \bar{\boldsymbol{X}}) \tilde{p}_{\boldsymbol{f},\boldsymbol{T},\boldsymbol{\lambda},\boldsymbol{U}}(\bar{\boldsymbol{X}}) d\bar{\boldsymbol{X}} = \int_{\mathbb{R}^d} p_{\boldsymbol{\varepsilon}}(\boldsymbol{X} - \bar{\boldsymbol{X}}) \tilde{p}_{\tilde{\boldsymbol{f}},\tilde{\boldsymbol{T}},\tilde{\boldsymbol{\lambda}},\tilde{\boldsymbol{U}}}(\bar{\boldsymbol{X}}) d\bar{\boldsymbol{X}} \qquad (27)$$
$$\implies (\tilde{p}_{\boldsymbol{f},\boldsymbol{T},\boldsymbol{\lambda},\boldsymbol{U}} * p_{\boldsymbol{\varepsilon}})(\boldsymbol{X}) = (\tilde{p}_{\tilde{\boldsymbol{f}},\tilde{\boldsymbol{T}},\tilde{\boldsymbol{\lambda}},\tilde{\boldsymbol{U}}} * p_{\boldsymbol{\varepsilon}})(\boldsymbol{X}) \qquad (28)$$
$$\implies F[\tilde{p}_{\boldsymbol{f},\boldsymbol{T},\boldsymbol{\lambda},\boldsymbol{U}}](\boldsymbol{\omega}) \varphi_{\boldsymbol{\varepsilon}}(\boldsymbol{\omega}) = F[\tilde{p}_{\tilde{\boldsymbol{f}},\tilde{\boldsymbol{T}},\tilde{\boldsymbol{\lambda}},\tilde{\boldsymbol{U}}}](\boldsymbol{\omega}) \varphi_{\boldsymbol{\varepsilon}}(\boldsymbol{\omega}) \qquad (29)$$
$$\implies F[\tilde{p}_{\boldsymbol{f},\boldsymbol{T},\boldsymbol{\lambda},\boldsymbol{U}}](\boldsymbol{\omega}) = F[\tilde{p}_{\tilde{\boldsymbol{f}},\tilde{\boldsymbol{T}},\tilde{\boldsymbol{\lambda}},\tilde{\boldsymbol{U}}}](\boldsymbol{\omega}) \qquad (30)$$
$$\implies \tilde{p}_{\boldsymbol{f},\boldsymbol{T},\boldsymbol{\lambda},\boldsymbol{U}}(\boldsymbol{X}) = \tilde{p}_{\tilde{\boldsymbol{f}},\tilde{\boldsymbol{T}},\tilde{\boldsymbol{\lambda}},\tilde{\boldsymbol{U}}}(\boldsymbol{X}) \qquad (31)$$
where

- in Eq. (26), $J$ denotes the Jacobian, and we made the change of variable $\bar{\boldsymbol{X}} = \boldsymbol{f}(\boldsymbol{Z})$ on the left hand side, and $\bar{\boldsymbol{X}} = \tilde{\boldsymbol{f}}(\boldsymbol{Z})$ on the right hand side.

- in Eq. (27), we introduced
$$\tilde{p}_{\boldsymbol{f},\boldsymbol{T},\boldsymbol{\lambda},\boldsymbol{U}}(\boldsymbol{X}) \triangleq p_{\boldsymbol{T},\boldsymbol{\lambda}}(\boldsymbol{f}^{-1}(\boldsymbol{X})|\boldsymbol{U}) \mathrm{vol}(J_{\boldsymbol{f}^{-1}}(\boldsymbol{X})) \mathbb{I}_{\mathcal{O}}(\boldsymbol{X}) \qquad (32)$$
on the left hand side, and similarly on the right hand side.

- in Eq. (28), we used $*$ for the convolution operator.

- in Eq. (29), we used $F[.]$ to designate the Fourier transform, and where $\varphi_{\boldsymbol{\varepsilon}} = F[p_{\boldsymbol{\varepsilon}}]$ (by definition of the characteristic function).

- in Eq. (30), we dropped $\varphi_{\boldsymbol{\varepsilon}}(\boldsymbol{\omega})$ from both sides as it is non-zero almost everywhere (by assumption (i)).

**Step II.** In this step, we remove all terms that are either a function of $\boldsymbol{X}$ or $\boldsymbol{U}$. First, by replacing both sides of Eq. (31) by their corresponding expressions from Eq. (32), we have
$$p_{\boldsymbol{T},\boldsymbol{\lambda}}(\boldsymbol{f}^{-1}(\boldsymbol{X})|\boldsymbol{U}) \mathrm{vol}(J_{\boldsymbol{f}^{-1}}(\boldsymbol{X})) = p_{\tilde{\boldsymbol{T}},\tilde{\boldsymbol{\lambda}}}(\tilde{\boldsymbol{f}}^{-1}(\boldsymbol{X})|\boldsymbol{U}) \mathrm{vol}(J_{\tilde{\boldsymbol{f}}^{-1}}(\boldsymbol{X})). \qquad (33)$$
Then, by taking logarithm on both sides of Eq. (33) and replacing $p_{\boldsymbol{T},\boldsymbol{\lambda}}$ by its expression from Eq. (21), we obtain
$$\log \mathrm{vol}(J_{\boldsymbol{f}^{-1}}(\boldsymbol{X})) + \log Q(\boldsymbol{f}^{-1}(\boldsymbol{X})) - \log Z(\boldsymbol{U}) + \left\langle \boldsymbol{T}(\boldsymbol{f}^{-1}(\boldsymbol{X})), \boldsymbol{\lambda}(\boldsymbol{U}) \right\rangle$$
$$= \log \mathrm{vol}(J_{\tilde{\boldsymbol{f}}^{-1}}(\boldsymbol{X})) + \log \tilde{Q}(\tilde{\boldsymbol{f}}^{-1}(\boldsymbol{X})) - \log \tilde{Z}(\boldsymbol{U}) + \left\langle \tilde{\boldsymbol{T}}(\tilde{\boldsymbol{f}}^{-1}(\boldsymbol{X})), \tilde{\boldsymbol{\lambda}}(\boldsymbol{U}) \right\rangle. \qquad (34)$$
Let $\boldsymbol{U}^0, \boldsymbol{U}^1, \cdots, \boldsymbol{U}^k \in \mathcal{U}$ be the $k+1$ points defined in assumption (iii). We evaluate the above equation at these points to obtain $k+1$ equations, and subtract the first equation from the remaining

$k$ equations to obtain:

$$\left\langle \boldsymbol{T}(\boldsymbol{f}^{-1}(\boldsymbol{X})), \boldsymbol{\lambda}(\boldsymbol{U}^l) - \boldsymbol{\lambda}(\boldsymbol{U}^0) \right\rangle + \log \frac{Z(\boldsymbol{U}^0)}{Z(\boldsymbol{U}^l)}$$

$$= \left\langle \tilde{\boldsymbol{T}}(\tilde{\boldsymbol{f}}^{-1}(\boldsymbol{X})), \tilde{\boldsymbol{\lambda}}(\boldsymbol{U}^l) - \tilde{\boldsymbol{\lambda}}(\boldsymbol{U}^0) \right\rangle + \log \frac{\tilde{Z}(\boldsymbol{U}^0)}{\tilde{Z}(\boldsymbol{U}^l)}, \quad l = 1, \cdots, k. \quad (35)$$

Let $L$ be defined as in assumption (iii) and $\tilde{L}$ defined similarly for $\tilde{\boldsymbol{\lambda}}$. Note that $L$ is invertible by assumption, but $\tilde{L}$ is not necessarily invertible. Letting $\boldsymbol{b} \in \mathbb{R}^k$ in which $b_l = \log \frac{\tilde{Z}(\boldsymbol{U}^0)Z(\boldsymbol{U}^l)}{\tilde{Z}(\boldsymbol{U}^l)Z(\boldsymbol{U}^0)}$, we have

$$L^T \boldsymbol{T}(\boldsymbol{f}^{-1}(\boldsymbol{X})) = \tilde{L}^T \tilde{\boldsymbol{T}}(\tilde{\boldsymbol{f}}^{-1}(\boldsymbol{X})) + \boldsymbol{b}. \quad (36)$$

Left multiplying both sides of the above equation by $L^{-T}$ gives

$$\boldsymbol{T}(\boldsymbol{f}^{-1}(\boldsymbol{X})) = A \tilde{\boldsymbol{T}}(\tilde{\boldsymbol{f}}^{-1}(\boldsymbol{X})) + \boldsymbol{c}, \quad (37)$$

where $A = L^{-T} \tilde{L} \in \mathbb{R}^{k \times k}$ and $\boldsymbol{c} = L^{-T} \boldsymbol{b} \in \mathbb{R}^k$.

**Step III.** To complete the proof, we need to show that $A$ is invertible. Let $\boldsymbol{Z}_l \in \mathcal{Z}$, $\boldsymbol{X}_l = \boldsymbol{f}(\boldsymbol{Z}_l)$, $l = 0, \cdots, k$. We evaluate Eq. (37) at these $k+1$ points to obtain $k+1$ equations and subtract the first equation from the remaining $k$ equations to obtain

$$\underbrace{[\boldsymbol{T}(\boldsymbol{Z}_1) - \boldsymbol{T}(\boldsymbol{Z}_0), \cdots, \boldsymbol{T}(\boldsymbol{Z}_k) - \boldsymbol{T}(\boldsymbol{Z}_0)]}_{\triangleq R \in \mathbb{R}^{k \times k}}$$

$$= A \underbrace{[\tilde{\boldsymbol{T}}(\tilde{\boldsymbol{f}}^{-1}(\boldsymbol{X}_1)) - \tilde{\boldsymbol{T}}(\tilde{\boldsymbol{f}}^{-1}(\boldsymbol{X}_0)), \cdots, \tilde{\boldsymbol{T}}(\tilde{\boldsymbol{f}}^{-1}(\boldsymbol{X}_k)) - \tilde{\boldsymbol{T}}(\tilde{\boldsymbol{f}}^{-1}(\boldsymbol{X}_0))]}_{\triangleq \tilde{R} \in \mathbb{R}^{k \times k}}. \quad (38)$$

We need to show that for a given $\boldsymbol{Z}_0 \in \mathcal{Z}$, there exist $k$ points $\boldsymbol{Z}_1, \cdots, \boldsymbol{Z}_k \in \mathcal{Z}$ such that the columns of $R$ are linearly independent. Suppose, for contradiction, that the columns of $R$ would never be linearly independent for any choice of $\boldsymbol{Z}_1, \cdots, \boldsymbol{Z}_k \in \mathcal{Z}$. Then the function $\mathbf{g}(\boldsymbol{Z}) \triangleq \boldsymbol{T}(\boldsymbol{Z}) - \boldsymbol{T}(\boldsymbol{Z}_0)$ would live in a $k-1$ or lower dimensional subspace, and thus we could find a non-zero vector $\boldsymbol{\lambda} \in \mathbb{R}^k$ orthogonal to that subspace. This would imply that $\langle \boldsymbol{T}(\boldsymbol{Z}) - \boldsymbol{T}(\boldsymbol{Z}_0), \boldsymbol{\lambda} \rangle = 0$ and thus $\langle \boldsymbol{T}(\boldsymbol{Z}), \boldsymbol{\lambda} \rangle = \langle \boldsymbol{T}(\boldsymbol{Z}_0), \boldsymbol{\lambda} \rangle = const, \forall \boldsymbol{Z} \in \mathcal{Z}$, which contradicts the assumption that the prior is strongly exponential (Definition 6). Therefore, we have shown that there exist $k+1$ points $\boldsymbol{Z}_0, \boldsymbol{Z}_1, \cdots, \boldsymbol{Z}_k \in \mathcal{Z}$ such that $R$ is invertible. Since $R = A\tilde{R}$ and $A$ is not a function of $\boldsymbol{Z}$, $A$ must be invertible. This completes the proof. $\square$

### H.1.2 PART II

**Theorem 5.** *Suppose that all assumptions in Theorem 4 hold. Let the sufficient statistics $\boldsymbol{T}(\boldsymbol{Z}) = [\boldsymbol{T}_f(\boldsymbol{Z})^T, \boldsymbol{T}_{NN}(\boldsymbol{Z})^T]^T$ given by the concatenation of a) the sufficient statistics $\boldsymbol{T}_f(\boldsymbol{Z}) = [\boldsymbol{T}_1(Z_1)^T, \cdots, \boldsymbol{T}_n(Z_n)^T]^T$ of a factorized exponential family, where all the $\boldsymbol{T}_i(Z_i)$ have dimension larger or equal to 2, and b) the output $\boldsymbol{T}_{NN}(\boldsymbol{Z})$ of a neural network with ReLU activations. (note that a neural network with ReLU activation has universal approximation power and should be able to capture any dependencies of interest). Let $k'$ be the dimension of $\boldsymbol{T}_f$ and thus that $k' \geq 2n$. Assume that*

*(i) the sufficient statistics $\boldsymbol{T}_f$ have all second-order own derivatives;*

*(ii) the mixing function $\boldsymbol{f}$ has all second-order cross derivatives.*

*Then the parameters $\{\boldsymbol{f}, \boldsymbol{T}, \boldsymbol{\lambda}\}$ are $\sim_P$ identifiable.*

*Proof.* Let $\boldsymbol{v} = \tilde{\boldsymbol{f}}^{-1} \circ \boldsymbol{f} : \mathcal{Z} \to \mathcal{Z}$. Since all assumptions in Theorem 4 hold, we have

$$\boldsymbol{T}(\boldsymbol{Z}) = A\tilde{\boldsymbol{T}}(\boldsymbol{v}(\boldsymbol{Z})) + \boldsymbol{c}, \quad (39)$$

where $A \in \mathbb{R}^{k \times k}$ is invertible. We want to show that $A$ is a block permutation matrix.

**Step I.** In this step, we show that $v$ is a componentwise function. First we differentiate both sides of Eq. (39) with respect to $Z_s$ and $Z_t$ ($s \neq t$) to obtain

$$\frac{\partial T(Z)}{\partial Z_s} = A \sum_{i=1}^{n} \frac{\partial \tilde{T}(v(Z))}{\partial v_i(Z)} \cdot \frac{\partial v_i(Z)}{\partial Z_s} \tag{40}$$

$$\frac{\partial^2 T(Z)}{\partial Z_s \partial Z_t} = A \sum_{i=1}^{n} \sum_{j=1}^{n} \frac{\partial^2 \tilde{T}(v(Z))}{\partial v_i(Z) \partial v_j(Z)} \cdot \frac{\partial v_j(Z)}{\partial Z_t} \cdot \frac{\partial v_i(Z)}{\partial Z_s} + A \sum_{i=1}^{n} \frac{\partial \tilde{T}(v(Z))}{\partial v_i(Z)} \cdot \frac{\partial^2 v_i(Z)}{\partial Z_s \partial Z_t}. \tag{41}$$

By construction, the second-order cross derivatives of $T$ and $\tilde{T}$ are all zero. Therefore, we have

$$0 = A \sum_{i=1}^{n} \frac{\partial^2 \tilde{T}(v(Z))}{\partial v_i(Z)^2} \cdot \frac{\partial v_i(Z)}{\partial Z_t} \cdot \frac{\partial v_i(Z)}{\partial Z_s} + A \sum_{i=1}^{n} \frac{\partial \tilde{T}(v(Z))}{\partial v_i(Z)} \cdot \frac{\partial^2 v_i(Z)}{\partial Z_s \partial Z_t}. \tag{42}$$

The above equation can be written in the matrix-vector form:

$$0 = A\tilde{T}''(Z)v'_{s,t}(Z) + A\tilde{T}'(Z)v''_{s,t}(Z), \tag{43}$$

where

$$\tilde{T}''(Z) = \left[\frac{\partial^2 \tilde{T}(v(Z))}{\partial v_1(Z)^2}, \cdots, \frac{\partial^2 \tilde{T}(v(Z))}{\partial v_n(Z)^2}\right] \in \mathbb{R}^{k \times n} \tag{44}$$

$$v'_{s,t}(Z) = \left[\frac{\partial v_1(Z)}{\partial Z_t} \cdot \frac{\partial v_1(Z)}{\partial Z_s}, \cdots, \frac{\partial v_n(Z)}{\partial Z_t} \cdot \frac{\partial v_n(Z)}{\partial Z_s}\right]^T \in \mathbb{R}^n, \tag{45}$$

and

$$\tilde{T}'(Z) = \left[\frac{\partial \tilde{T}(v(Z))}{\partial v_1(Z)}, \cdots, \frac{\partial \tilde{T}(v(Z))}{\partial v_n(Z)}\right] \in \mathbb{R}^{k \times n} \tag{46}$$

$$v''_{s,t}(Z) = \left[\frac{\partial^2 v_1(Z)}{\partial Z_s \partial Z_t}, \cdots, \frac{\partial^2 v_n(Z)}{\partial Z_s \partial Z_t}\right]^T \in \mathbb{R}^n. \tag{47}$$

Now by concatenating

$$\tilde{T}'''(Z) = [\tilde{T}''(Z), \tilde{T}'(Z)] \in \mathbb{R}^{k \times 2n} \quad \text{and} \quad v'''_{s,t}(Z) = [v'_{s,t}(Z)^T, v''_{s,t}(Z)^T]^T \in \mathbb{R}^{2n}, \tag{48}$$

we obtain

$$0 = A\tilde{T}'''(Z)v'''_{s,t}(Z). \tag{49}$$

Finally, we take the rows of $\tilde{T}'''(Z)$ that corresponds to the factorized strongly exponential family distribution part and denote them by $\tilde{T}'''_f(Z) \in \mathbb{R}^{k' \times 2n}$. By Lemma 5 in the iVAE paper (Khemakhem et al., 2020a) and the assumption that $k' \geq 2n$, we have that the rank of $\tilde{T}'''_f(Z)$ is $2n$. Since $k \geq k' \geq 2n$, the rank of $\tilde{T}'''(Z)$ is also $2n$. Since the rank of $A$ is $k$, the rank of $A\tilde{T}'''(Z) \in \mathbb{R}^{k \times 2n}$ is $2n$. This implies that $v'''_{s,t}(Z)$ must be a zero vector. In particular, we have that $v'_{s,t}(Z) = 0$, $\forall s \neq t$. Therefore, we have shown that $v$ is a componentwise function.

**Step II.** To complete the proof, we need to show that $A$ is a block permutation matrix. Without loss of generality, we assume that the permutation in $v$ is the identity. That is $v(Z) = [v_1(Z_1), \cdots, v_n(Z_n)]^T$ for some nonlinear univariate scalar functions $v_1, \cdots, v_n$. Since $f$ and $\tilde{f}$ are bijective, we have that $v$ is also bijective and $v^{-1}(Z) = [v_1^{-1}(Z_1), \cdots, v_n^{-1}(Z_n)]^T$. We denote $\bar{T}(v(Z)) = \tilde{T}(v(Z)) + A^{-1}c$ and plug it into Eq. (39) to obtain $T(Z) = A\bar{T}(v(Z))$. Applying $v^{-1}$ to the variables $Z$ at both sides gives

$$T(v^{-1}(Z)) = A\bar{T}(Z). \tag{50}$$

Let $t$ be the index of an entry in the sufficient statistics $T$ that corresponds to the the factorized strongly exponential family distribution part $T_f$. For all $s \neq t$, we have

$$0 = \frac{\partial T(v^{-1}(Z))_t}{\partial Z_s} = \sum_{j=1}^{k} a_{tj} \frac{\partial \bar{T}(Z)_j}{\partial Z_s}. \tag{51}$$

Since the entries of $\tilde{T}$ are linearly independent (if they were not linearly independent, then $\tilde{T}$ can be compressed into a smaller vector by removing the redundant entries), we have that $a_{tj}$ is zero for any $j$ such that $\frac{\partial \bar{T}(Z)_j}{\partial Z_s} \neq 0$. This includes the entries $j$ in the sufficient statistics $\tilde{T}$ that corresponds to 1) the factorized strongly exponential family distribution part which do not depend on $Z_t$; and 2) the neural network part.

Therefore, when $t$ is the index of an entry in the sufficient statistics $\boldsymbol{T}$ that corresponds to factor $i$ in the factorized strongly exponential family distribution part $\boldsymbol{T}_f$, the only non-zero $a_{tj}$ are the ones that map between $\boldsymbol{T}_i(Z_i)$ and $\bar{\boldsymbol{T}}_i(v_i(Z_i))$, where $\boldsymbol{T}_i$ are the factors in $\boldsymbol{T}_f$ that only depend on $Z_i$ and $\bar{\boldsymbol{T}}_i$ is defined similarly. Therefore, we can construct an invertible submatrix $A_i'$ with all non-zero elements $a_{tj}$ for all $t$ that corresponds to factor $i$, such that

$$\boldsymbol{T}_i(Z_i) = A_i'\bar{\boldsymbol{T}}_i(v_i(Z_i)) = A_i'\tilde{\boldsymbol{T}}_i(v_i(Z_i)) + \boldsymbol{c}_i, \quad i = 1, \cdots, n, \tag{52}$$

where $\tilde{\boldsymbol{T}}_i$ are the factors in $\tilde{\boldsymbol{T}}_f$ that only depends on $Z_i$, and $\boldsymbol{c}_i$ are the corresponding elements of $\boldsymbol{c}$. This means that the matrix $A$ is a block permutation matrix. For each $i = 1, \cdots, n$, the block $A_i'$ of $A$ affinely transforms $\boldsymbol{T}_i(Z_i)$ into $\tilde{\boldsymbol{T}}_i(v_i(Z_i))$. There is also an additional block $A'$ which affinely transforms $\boldsymbol{T}_{NN}(\boldsymbol{Z})$ into $\tilde{\boldsymbol{T}}_{NN}(\boldsymbol{v}(\boldsymbol{Z}))$. This completes the proof. □

### H.1.3 PART III

Now we combine Theorem 4 in Part I and Theorem 5 in Part II into one theorem, which completes the proof of Theorem 1.

### H.2 PROOF OF THEOREM 2

We recall that the loss function in Phase 1 is as follows:
$$\mathcal{L}_{\text{phase1}}(\boldsymbol{\theta}, \boldsymbol{\phi}) = \mathcal{L}_{\text{phase1}}^{\text{ELBO}}(\boldsymbol{f}, \hat{\boldsymbol{T}}, \hat{\boldsymbol{\lambda}}, \boldsymbol{\phi}) - \mathcal{L}_{\text{phase1}}^{\text{SM}}(\hat{\boldsymbol{f}}, \boldsymbol{T}, \boldsymbol{\lambda}, \hat{\boldsymbol{\phi}}), \tag{53}$$
where
$$\mathcal{L}_{\text{phase1}}^{\text{ELBO}}(\boldsymbol{\theta}, \boldsymbol{\phi}) := \mathbb{E}_{p_D}\left[\mathbb{E}_{q_{\boldsymbol{\phi}}(\boldsymbol{Z}|\boldsymbol{X}, \boldsymbol{Y}, \boldsymbol{E})}\left[\log p_{\boldsymbol{f}}(\boldsymbol{X}|\boldsymbol{Z}) + \log p_{\boldsymbol{T}, \boldsymbol{\lambda}}(\boldsymbol{Z}|\boldsymbol{Y}, \boldsymbol{E}) - \log q_{\boldsymbol{\phi}}(\boldsymbol{Z}|\boldsymbol{X}, \boldsymbol{Y}, \boldsymbol{E})\right]\right], \tag{54}$$
$$\mathcal{L}_{\text{phase1}}^{\text{SM}}(\boldsymbol{T}, \boldsymbol{\lambda}) := \mathbb{E}_{p_D}\left[\mathbb{E}_{q_{\boldsymbol{\phi}}(\boldsymbol{Z}|\boldsymbol{X}, \boldsymbol{Y}, \boldsymbol{E})}\left[||\nabla_{\boldsymbol{Z}} \log q_{\boldsymbol{\phi}}(\boldsymbol{Z}|\boldsymbol{X}, \boldsymbol{Y}, \boldsymbol{E}) - \nabla_{\boldsymbol{Z}} \log p_{\boldsymbol{T}, \boldsymbol{\lambda}}(\boldsymbol{Z}|\boldsymbol{Y}, \boldsymbol{E})||^2\right]\right]. \tag{55}$$

*Proof.* If the family of $q_{\boldsymbol{\phi}}(\boldsymbol{Z}|\boldsymbol{X}, \boldsymbol{Y}, \boldsymbol{E})$ is flexible enough to contain $p_{\boldsymbol{\theta}}(\boldsymbol{Z}|\boldsymbol{X}, \boldsymbol{Y}, \boldsymbol{E})$, then by optimizing the loss over its parameter $\boldsymbol{\phi}$, we will minimize the score matching term $\mathcal{L}_{\text{phase1}}^{\text{SM}}$ in Eq. (55), which will eventually reach zero. If we assume that the model is not degenerate and that $q_{\boldsymbol{\phi}} > 0$ everywhere, then having that $\mathcal{L}_{\text{phase1}}^{\text{SM}} = 0$ implies that $\nabla_{\boldsymbol{Z}} \log q_{\boldsymbol{\phi}}(\boldsymbol{Z}|\boldsymbol{X}, \boldsymbol{Y}, \boldsymbol{E})$ and $\nabla_{\boldsymbol{Z}} \log p_{\boldsymbol{T}, \boldsymbol{\lambda}}(\boldsymbol{Z}|\boldsymbol{Y}, \boldsymbol{E})$ are equal. This implies that $\log q_{\boldsymbol{\phi}}(\boldsymbol{Z}|\boldsymbol{X}, \boldsymbol{Y}, \boldsymbol{E}) = \log p_{\boldsymbol{T}, \boldsymbol{\lambda}}(\boldsymbol{Z}|\boldsymbol{Y}, \boldsymbol{E}) + c$ for some constant $c$. But $c$ is necessarily 0 because both $q_{\boldsymbol{\phi}}(\boldsymbol{Z}|\boldsymbol{X}, \boldsymbol{Y}, \boldsymbol{E})$ and $p_{\boldsymbol{T}, \boldsymbol{\lambda}}(\boldsymbol{Z}|\boldsymbol{Y}, \boldsymbol{E})$ are pdf's. Therefore, the ELBO term $\mathcal{L}_{\text{phase1}}^{\text{ELBO}}$ in Eq. (54) will be equal to the log-likelihood, meaning that the loss $\mathcal{L}_{\text{phase1}}$ in Eq. (53) will be equal to the log-likelihood. Under this circumstance, the estimation in Eq. (53) inherits all the properties of maximum likelihood estimation (MLE). In this particular case, since our identifiability is guaranteed up to a permutation and componentwise transformation, the consistency of MLE means that we converge to the true parameter $\boldsymbol{\theta}^*$ up to a permutation and componentwise transformation in the limit of infinite data. Because true identifiability is one of the assumptions for MLE consistency, replacing it by identifiability up to a permutation and componentwise transformation does not change the proof but only the conclusion. □

### H.3 PROOF OF THEOREM 3

*Proof.* Theorem 1 and Theorem 2 guarantee that in the limit of infinite data, NF-iVAE can learn the true parameters $\boldsymbol{\theta}^* := (\boldsymbol{f}^*, \boldsymbol{T}^*, \boldsymbol{\lambda}^*)$ up to a permutation and componentwise transformation of the latent variables. Let $(\hat{\boldsymbol{f}}, \hat{\boldsymbol{T}}, \hat{\boldsymbol{\lambda}})$ be the parameters obtained by NF-iVAE. We, therefore, have $(\hat{\boldsymbol{f}}, \hat{\boldsymbol{T}}, \hat{\boldsymbol{\lambda}}) \sim_P (\boldsymbol{f}^*, \boldsymbol{T}^*, \boldsymbol{\lambda}^*)$, where $\sim_P$ denotes the equivalence up to a permutation and componentwise transformation. If there were no noise, this would mean that the learned $\hat{\boldsymbol{f}}$ transforms $\boldsymbol{X}$ into $\hat{\boldsymbol{Z}} = \hat{\boldsymbol{f}}^{-1}(\boldsymbol{X})$ that are equal to $\boldsymbol{Z}^* = (\boldsymbol{f}^*)^{-1}(\boldsymbol{X})$ up to a permutation and componentwise transformation (Definition 9). If with noise, we obtain the posterior distribution of the latent variables up to an analogous indeterminacy. □

### H.4 PROOF OF PROPOSITION 1

*Proof.* Theorem A.1 in (Arjovsky, 2021) has showed that i) any predictor $w \circ \Phi$ with optimal OOD generalization uses only $\text{Pa}(\boldsymbol{Y})$ to compute $\Phi$; ii) the classifier $w$ in this optimal predictor can be estimated using data from any environment $e$ for which the distribution of $\text{Pa}(\boldsymbol{Y}^e)$ has full support;

iii) the optimal predictor will be invariant across $\mathcal{E}_{all}$. In iCaRL, the hypotheses of Theorems 1 and 2 and Assumption 1 guarantee that $\mathrm{Pa}(\boldsymbol{Y})$ can be recovered by first identifying the latent variables $\boldsymbol{Z}$ from $\boldsymbol{X}, \boldsymbol{Y}$ and $\boldsymbol{E}$ and then discovering the direct causes of $\boldsymbol{Y}$ through solving Eq. (12). Furthermore, since the conditional prior in Eq. (5) of the main text has full support, the distribution of $\mathrm{Pa}(\boldsymbol{Y}^e)$ always has full support. Also, under Assumption 1 we have that $p(\boldsymbol{Y}|\mathrm{Pa}(\boldsymbol{Y}))$ is invariant across $\mathcal{E}_{all}$. Hence, the classifier $w$ in this optimal predictor can be estimated using data from any environment $e$. We therefore have that the resulting optimal predictor will be invariant across $\mathcal{E}_{all}$. This completes the proof.

$\square$

## H.5    Proof of Proposition 2

*Proof.* The following rules can be independently performed to distinguish all the 10 structures shown in Figs. 3b-3k. For clarity, we divide them into three groups. Note that, since these rules rely on different algorithms of causal discovery and conditional independence tests, unless stated otherwise, we assume that the assumptions of these algorithms are satisfied during the proof process.

**Group 1**    All the six structures in this group can be discovered only by performing conditional independence tests.

- **Rule 1.1** If $Z_i \perp\!\!\!\perp \boldsymbol{Y}$, $Z_i \not\perp\!\!\!\perp \boldsymbol{E}$, and $\boldsymbol{E} \perp\!\!\!\perp \boldsymbol{Y}$, then Fig. 3d is discovered.

- **Rule 1.2** If $Z_i \not\perp\!\!\!\perp \boldsymbol{Y}$, $Z_i \perp\!\!\!\perp \boldsymbol{E}$, and $\boldsymbol{E} \not\perp\!\!\!\perp \boldsymbol{Y}$, then Fig. 3g is discovered.

- **Rule 1.3** If $Z_i \not\perp\!\!\!\perp \boldsymbol{Y}$, $Z_i \not\perp\!\!\!\perp \boldsymbol{E}$, and $\boldsymbol{E} \perp\!\!\!\perp \boldsymbol{Y}$, then Fig. 3f is discovered.

- **Rule 1.4** If $Z_i \not\perp\!\!\!\perp \boldsymbol{Y}$, $Z_i \not\perp\!\!\!\perp \boldsymbol{E}$, $\boldsymbol{E} \not\perp\!\!\!\perp \boldsymbol{Y}$, and $Z_i \perp\!\!\!\perp \boldsymbol{Y}|\boldsymbol{E}$, then Fig. 3i is discovered.

- **Rule 1.5** If $Z_i \not\perp\!\!\!\perp \boldsymbol{Y}$, $Z_i \not\perp\!\!\!\perp \boldsymbol{E}$, $\boldsymbol{E} \not\perp\!\!\!\perp \boldsymbol{Y}$, and $Z_i \perp\!\!\!\perp \boldsymbol{E}|\boldsymbol{Y}$, then Fig. 3h is discovered.

- **Rule 1.6** If $Z_i \not\perp\!\!\!\perp \boldsymbol{Y}$, $Z_i \not\perp\!\!\!\perp \boldsymbol{E}$, $\boldsymbol{E} \not\perp\!\!\!\perp \boldsymbol{Y}$, and $\boldsymbol{Y} \perp\!\!\!\perp \boldsymbol{E}|Z_i$, then Fig. 3e is discovered.

**Group 2**    If $Z_i \not\perp\!\!\!\perp \boldsymbol{Y}$, $Z_i \perp\!\!\!\perp \boldsymbol{E}$, and $\boldsymbol{E} \perp\!\!\!\perp \boldsymbol{Y}$, then we can discover both Fig. 3b and Fig. 3c. These two structures cannot be further distinguished only by conditional independence tests, because they come from the same Markov equivalence class. Fortunately, we can further distinguish them by running binary causal discovery algorithms (Peters et al., 2017), e.g., ANM (Hoyer et al., 2009) for continuous data and CDS (Fonollosa, 2019) for continuous and discrete data.

- **Rule 2.1** If $Z_i \not\perp\!\!\!\perp \boldsymbol{Y}$, $Z_i \perp\!\!\!\perp \boldsymbol{E}$, and $\boldsymbol{E} \perp\!\!\!\perp \boldsymbol{Y}$, and a chosen binary causal discovery algorithm prefers $Z_i \to \boldsymbol{Y}$ to $Z_i \leftarrow \boldsymbol{Y}$, then Fig. 3b is discovered.

- **Rule 2.2** If $Z_i \not\perp\!\!\!\perp \boldsymbol{Y}$, $Z_i \perp\!\!\!\perp \boldsymbol{E}$, and $\boldsymbol{E} \perp\!\!\!\perp \boldsymbol{Y}$, and a chosen binary causal discovery algorithm prefers $Z_i \leftarrow \boldsymbol{Y}$ to $Z_i \to \boldsymbol{Y}$, then Fig. 3c is discovered.

**Group 3**    If $Z_i \not\perp\!\!\!\perp \boldsymbol{Y}$, $Z_i \not\perp\!\!\!\perp \boldsymbol{E}$, $\boldsymbol{E} \not\perp\!\!\!\perp \boldsymbol{Y}$, $Z_i \not\perp\!\!\!\perp \boldsymbol{Y}|\boldsymbol{E}$, $Z_i \not\perp\!\!\!\perp \boldsymbol{E}|\boldsymbol{Y}$, and $\boldsymbol{Y} \not\perp\!\!\!\perp \boldsymbol{E}|Z_i$, then we can discover both Fig. 3j and Fig. 3k. These two structures cannot be further distinguished only by conditional independence tests, because they come from the same Markov equivalence class. They also cannot be distinguished by any binary causal discovery algorithm, since both $Z_i$ and $\boldsymbol{Y}$ are affected by $\boldsymbol{E}$. Fortunately, Zhang et al. (2017) provided a heuristic solution to this case based on the invariance of causal mechanisms, i.e., $P(\text{cause})$ and $P(\text{effect}|\text{cause})$ change independently. The detailed description of their method is given in Section 4.2 of Zhang et al. (2017). For convenience, here we directly borrow their final result. Zhang et al. (2017) states that determining the causal direction between $Z_i$ and $\boldsymbol{Y}$ in Fig. 3j and Fig. 3k is finally reduced to calculating the following term:

$$\Delta_{Z_i \to \boldsymbol{Y}} = \left\langle \log \frac{\bar{P}(\boldsymbol{Y}|Z_i)}{\langle \hat{P}(\boldsymbol{Y}|Z_i) \rangle} \right\rangle, \tag{56}$$

where $\langle \cdot \rangle$ denotes the sample average, $\bar{P}(\boldsymbol{Y}|Z_i)$ is the empirical estimate of $P(\boldsymbol{Y}|Z_i)$ on all data points, and $\langle \hat{P}(\boldsymbol{Y}|Z_i) \rangle$ denotes the sample average of $\hat{P}(\boldsymbol{Y}|Z_i)$, which is the estimate of $P(\boldsymbol{Y}|Z_i)$ in each environment. We take the direction for which $\Delta$ is smaller to be the causal direction.

- **Rule 3.1** If $Z_i \not\perp\!\!\!\perp Y$, $Z_i \not\perp\!\!\!\perp E$, $E \not\perp\!\!\!\perp Y$, $Z_i \not\perp\!\!\!\perp Y|E$, $Z_i \not\perp\!\!\!\perp E|Y$, $Y \not\perp\!\!\!\perp E|Z_i$, and $\Delta_{Z_i \to Y}$ is smaller than $\Delta_{Y \to Z_i}$, then Fig. 3j is discovered.

- **Rule 3.2** If $Z_i \not\perp\!\!\!\perp Y$, $Z_i \not\perp\!\!\!\perp E$, $E \not\perp\!\!\!\perp Y$, $Z_i \not\perp\!\!\!\perp Y|E$, $Z_i \not\perp\!\!\!\perp E|Y$, $Y \not\perp\!\!\!\perp E|Z_i$, and $\Delta_{Y \to Z_i}$ is smaller than $\Delta_{Z_i \to Y}$, then Fig. 3k is discovered.

$\square$

# I    ILLUSTRATIONS FOR MODEL LEARNING

As described in Section 3.3, in the ground truth model, the prior for each $Z_{i \in I_p}$ is either $p(Z_i|\boldsymbol{E})$ or $p(Z_i)$, depending on whether $Z_i$ is caused by $\boldsymbol{E}$ or not. By contrast, in the NF-iVAE model the prior is $p(\boldsymbol{Z}|\boldsymbol{Y}, \boldsymbol{E})$. Is this going to affect the identifiability of the latent variables? Well, in practice not because the posterior distribution for $\boldsymbol{Z}$ given $\boldsymbol{X}$, $\boldsymbol{Y}$ and $\boldsymbol{E}$ would be equivalent in both models (up to identifiability guarantees).

# J    DATASETS

For convenience and completeness, we provide descriptions of Colored MNIST Digits and Colored Fashion MNIST here. Please refer to the original papers (Arjovsky et al., 2019; Ahuja et al., 2020a; Gulrajani & Lopez-Paz, 2020; Venkateswara et al., 2017) for more details.

## J.1    SYNTHETIC DATA

For the nonlinear transformation, we use the MLP:

- Input layer: Input batch *(batch size, input dimension)*
- Layer 1: Fully connected layer, output size = 6, activation = ReLU
- Output layer: Fully connected layer, output size = 10

## J.2    COLORED MNIST DIGITS

We use the exact same environment as in Arjovsky et al. (2019). Arjovsky et al. (2019) propose to create an environment for training to classify digits in MNIST data[10], where the images in MNIST are now colored in such a way that the colors spuriously correlate with the labels. The task is to classify whether the digit is less than 5 (not including 5). There are three environments (two training containing 30,000 points each, one test containing 10,000 points) We add noise to the preliminary label ($\tilde{y} = 0$ if the digit is between 0-4 and $\tilde{y} = 1$ if the digit is between 5-9) by flipping it with 25 percent probability to construct the final labels. We sample the color id $z$ by flipping the final labels with probability $p_e$, where $p_e$ is 0.2 in the first environment, 0.1 in the second environment, and 0.9 in the third environment. The third environment is the testing environment. We color the digit red if $z = 1$ or green if $z = 0$.

## J.3    COLORED FASHION MNIST

We modify the fashion MNIST dataset[11] in a manner similar to the MNIST digits dataset. Fashion MNIST data has images from different categories: "t-shirt", "trouser", "pullover", "dress", "coat", "sandal", "shirt", "sneaker", "bag", "ankle boots". We add colors to the images in such a way that the colors correlate with the labels. The task is to classify whether the image is that of foot wear or a clothing item. There are three environments (two training, one test) We add noise to the preliminary label ($\tilde{y} = 0$: "t-shirt", "trouser", "pullover", "dress", "coat", "shirt" and $\tilde{y} = 1$: "sandal", "sneaker", "ankle boots") by flipping it with 25 percent probability to construct the final label. We sample the color id $z$ by flipping the noisy label with probability $p_e$, where $p_e$ is 0.2 in the first environment, 0.1

---

[10] https://www.tensorflow.org/api_docs/python/tf/keras/datasets/mnist/load_data

[11] https://www.tensorflow.org/api_docs/python/tf/keras/datasets/fashion_mnist/load_data

Table 3: Results on synthetic data. Comparisons are in terms of MSE (mean $\pm$ std deviation).

| $g(\cdot)$ | METHOD | TRAIN $(\sigma_3 = \{0.2, 2\})$ | TEST $(\sigma_3 = 100)$ |
|---|---|---|---|
| Identity | ERM | $0.00 \pm 0.00$ | $\mathbf{0.00 \pm 0.00}$ |
| | IRM | $0.00 \pm 0.00$ | $\mathbf{0.00 \pm 0.00}$ |
| | F-IRM GAME | $0.98 \pm 0.23$ | $1.03 \pm 0.04$ |
| | V-IRM GAME | $0.99 \pm 2.74$ | $1.07 \pm 2.26$ |
| | **iCaRL (ours)** | $0.01 \pm 0.03$ | $1.00 \pm 0.01$ |
| Linear | ERM | $0.00 \pm 0.00$ | $\mathbf{0.00 \pm 0.00}$ |
| | IRM | $0.00 \pm 0.00$ | $\mathbf{0.00 \pm 0.00}$ |
| | F-IRM GAME | $0.99 \pm 0.01$ | $1.08 \pm 0.06$ |
| | V-IRM GAME | $1.00 \pm 5.98$ | $1.05 \pm 0.04$ |
| | **iCaRL (ours)** | $0.01 \pm 0.03$ | $1.01 \pm 0.04$ |
| Nonlinear | ERM | $0.06 \pm 0.01$ | $220.79 \pm 229.97$ |
| | IRM | $0.08 \pm 0.01$ | $149.60 \pm 104.85$ |
| | F-IRM GAME | $1.06 \pm 0.09$ | $196.59 \pm 150.71$ |
| | V-IRM GAME | $1.00 \pm 0.01$ | $170.46 \pm 125.62$ |
| | **iCaRL (ours)** | $0.29 \pm 0.04$ | $\mathbf{28.16 \pm 2.54}$ |

in the second environment, and 0.9 in the third environment, which is the test environment. We color the object red if $z = 1$ or green if $z = 0$.

## K    IN-DEPTH ANALYSIS ON SYNTHETIC DATA

### K.1    COMPARISONS WITH STATE-OF-THE-ART

We first conduct a series of experiments on synthetic data generated according to an extension of the SEM in Model 1. The extension is to map the variables $\boldsymbol{Z} := (Z_1, Z_2)$ into a 10 dimensional observation $\boldsymbol{X}$ through a linear or nonlinear transformation. Our goal is to predict $Y$ from $\boldsymbol{X}$, where $\boldsymbol{X} = g(\boldsymbol{Z})$. We consider three transformations: (a) *Identity*: $g(\cdot)$ is the identity matrix $\boldsymbol{I} \in \mathbb{R}^{2\times2}$, i.e., $\boldsymbol{X} = g(\boldsymbol{Z}) = \boldsymbol{Z}$. (b) *Linear*: $g(\cdot)$ is a random matrix $\boldsymbol{S} \in \mathbb{R}^{2\times10}$, i.e., $\boldsymbol{X} = g(\boldsymbol{Z}) = \boldsymbol{Z} \cdot \boldsymbol{S}$. (c) *Nonlinear*: $g(\cdot)$ is given by a neural network with 2-dimensional input and 10-dimensional output, whose parameters are randomly set in advance. Since this is a regression task, we use the mean squared error (MSE) as a metric of performance. Note that, in this problem, there is only one causal latent variable $Z_1$, meaning that the conditional prior in Eq. (5) will not exhibit dependencies. Because of this, in this case we do not include a $\boldsymbol{T}_{NN}(\boldsymbol{Z})$ term in our NF-iVAE conditional prior. In the following section we do consider settings with many potential causal latent variables and, in that case, we do include $\boldsymbol{T}_{NN}(\boldsymbol{Z})$ in the NF-iVAE prior.

We consider a simple scenario in which we fix $\sigma_1 = 1$ and $\sigma_2 = 0$ for all environments and only allow $\sigma_3$ to vary across environments. In this case, $\sigma_3$ controls the spurious correlations between $Z_2$ and $Y$. Each experiment draws 1000 samples from each of the three environments $\sigma_3 = \{0.2, 2, 100\}$, where the first two are for training and the third for testing. We compare with the following baselines:[12] ERM, and two variants of IRMG: F-IRM Game (with $\Phi$ fixed to the identity) and V-IRM Game (with variable $\Phi$).

As shown in Table 3, in the cases of *Identity* and *Linear*, our approach is better than IRMG but only comparable with ERM and IRM. This might be because the identifiability result up to a simple nonlinear transformation renders the problem more difficult by converting the original identity or linear problem into a nonlinear problem. In the *Nonlinear* case, the gains of iCaRL are very clear.

Table 4: Comparisons of assumptions on the prior leading to identifiability in different algorithms.

| METHOD | Assumption on the Prior for Identifiability |
|---|---|
| VAE (Kingma & Welling, 2013) | Non-identifiability with $p_{\boldsymbol{T}, \boldsymbol{\lambda}}(\boldsymbol{Z}) = \prod_i p(Z_i) \overset{e.g.}{=} \mathcal{N}(\boldsymbol{0}, \boldsymbol{I})$ |
| iVAE (Khemakhem et al., 2020a) | $p_{\boldsymbol{T}, \boldsymbol{\lambda}}(\boldsymbol{Z}|\boldsymbol{Y}, \boldsymbol{E}) = \prod_i \mathcal{Q}_i(Z_i)/\mathcal{C}_i(\boldsymbol{Y}, \boldsymbol{E}) \exp[\sum_{j=1}^{k} T_{i,j}(Z_i)\lambda_{i,j}(\boldsymbol{Y}, \boldsymbol{E})]$ |
| ICE-BeeM (Khemakhem et al., 2020b) | $p_{\boldsymbol{T}, \boldsymbol{\lambda}}(\boldsymbol{Z}|\boldsymbol{Y}, \boldsymbol{E}) = \mathcal{Q}(\boldsymbol{Z})/\mathcal{C}(\boldsymbol{Y}, \boldsymbol{E}) \prod_i \exp[\sum_{j=1}^{k} T_{i,j}(Z_i)\lambda_{i,j}(\boldsymbol{Y}, \boldsymbol{E})]$ |
| NF-iVAE | $p_{\boldsymbol{T}, \boldsymbol{\lambda}}(\boldsymbol{Z}|\boldsymbol{Y}, \boldsymbol{E}) = \mathcal{Q}(\boldsymbol{Z})/\mathcal{C}(\boldsymbol{Y}, \boldsymbol{E}) \exp[\boldsymbol{T}(\boldsymbol{Z})^{\mathrm{T}} \boldsymbol{\lambda}(\boldsymbol{Y}, \boldsymbol{E})]$ |

---

[12]We also tried ICP, but ICP was unable to find any parent of $Y$ even in the identity case.

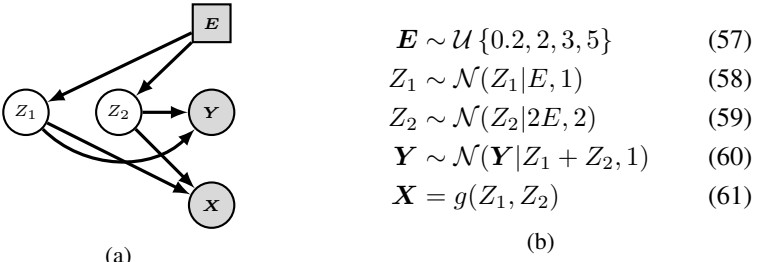

$$E \sim \mathcal{U}\{0.2, 2, 3, 5\} \qquad (57)$$
$$Z_1 \sim \mathcal{N}(Z_1|E, 1) \qquad (58)$$
$$Z_2 \sim \mathcal{N}(Z_2|2E, 2) \qquad (59)$$
$$\mathbf{Y} \sim \mathcal{N}(\mathbf{Y}|Z_1 + Z_2, 1) \qquad (60)$$
$$\mathbf{X} = g(Z_1, Z_2) \qquad (61)$$

(a)            (b)

Figure 4: (a) Causal structure with $\mathbf{Y}$ having two causes. (b) Data generating process corresponding to (a), where $\mathcal{U}\{\cdot\}$ denotes the discrete uniform distribution, $\mathcal{N}(\cdot)$ the Gaussian distribution, and $g(\cdot)$ is given by a neural network with 2-dimensional input and 10-dimensional output, whose parameters are randomly set in advance.

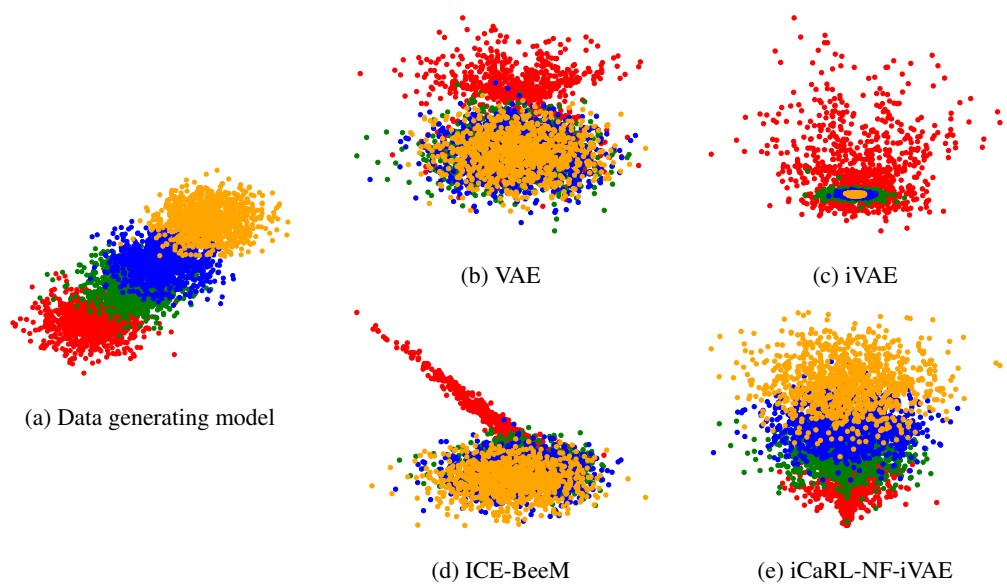

Figure 5: Visualization of the samples (i.e., $\hat{\mathbf{Z}} = (\hat{Z}_1, \hat{Z}_2)$) in latent space generated through different algorithms. (a) Samples from the true distribution. (b-e) Samples from the posterior of different algorithms. Apparently, our method (e) can recover the original data up to a permutation and a simple componentwise transformation.

## K.2    VISUALIZATION OF IDENTIFIABILITY OF NF-IVAE

To further verify identifiability of NF-iVAE, we conduct a series of experiments on synthetic data generated according to the data generating process (Fig. 4b) corresponding to the causal graph shown in Fig. 4a. The reason we choose this setting is that it is the simplest case satisfying our requirements: a) For ease of visualization, the latent space had better be 2-dimensional; b) To introduce the non-factorized prior given $\mathbf{Y}$ and $\mathbf{E}$ (i.e., $Z_i \not\perp\!\!\!\perp Z_j | \mathbf{Y}, \mathbf{E}$), $\mathbf{Y}$ has at least two causes.

We draw 1000 samples from each of the four environments $\mathbf{E} = \{0.2, 2, 3, 5\}$, and thus the whole synthetic dataset consists of 4000 samples. We compare with the following baselines: VAE Kingma & Welling (2013) (without identifiability guarantees), iVAE Khemakhem et al. (2020a) (with a conditionally factorized prior for identifiability), and ICE-BeeM Khemakhem et al. (2020b). It is worth noting that in ICE-BeeM the primary assumption leading to identifiability is similar to that in iVAE, where the base measure $\mathcal{Q}(\mathbf{Z})$ could be arbitrary to capture the dependences between latent variables but the exponential term still has to factorize across components (dimensions). All these are summarized in Table 4. Clearly, from the table we can see that our method has a more general assumption on the prior leading to identifiability. This is also demonstrated empirically in Fig. 5. Our method iCaRL can recover the original data $\mathbf{Z}$ up to a permutation and a simple componentwise

Table 5: Colored MNIST. Comparisons in terms of accuracy (%) (mean ± std deviation).

| METHOD | TRAIN | TEST |
|---|---|---|
| ERM | 84.88 ± 0.16 | 10.45 ± 0.66 |
| ERM 1 | 84.84 ± 0.21 | 10.86 ± 0.52 |
| ERM 2 | 84.95 ± 0.20 | 10.05 ± 0.23 |
| ROBUST MIN MAX | 84.25 ± 0.43 | 15.24 ± 2.45 |
| F-IRM GAME | 63.37 ± 1.14 | 59.91 ± 2.69 |
| V-IRM GAME | 63.97 ± 1.03 | 49.06 ± 3.43 |
| IRM | 59.27 ± 4.39 | 62.75 ± 9.59 |
| **iCaRL (ours)** | **70.56 ± 0.81** | **68.75 ± 1.45** |
| ERM GRAYSCALE | 71.81 ± 0.47 | 71.36 ± 0.65 |
| OPTIMAL | 75 | 75 |

Table 6: PACS. Comparisons in terms of accuracy (%) (mean ± std deviation).

| METHOD | TEST |
|---|---|
| ERM | 85.7 ± 0.5 |
| IRM | 84.4 ± 1.1 |
| DRO (Sagawa et al., 2019) | 84.1 ± 0.4 |
| Mixup (Yan et al., 2020) | 84.3 ± 0.5 |
| CORAL (Sun & Saenko, 2016) | 86.0 ± 0.2 |
| MMD (Li et al., 2018b) | 85.0 ± 0.2 |
| DANN (Ganin et al., 2016) | 84.6 ± 1.1 |
| C-DANN (Li et al., 2018c) | 82.8 ± 1.5 |
| LaCIM (Sun et al., 2020) | 83.5 ± 1.2 |
| **iCaRL (ours)** | **88.7 ± 0.6** |

transformation, whereas all the other methods fail because they are unable to handle the case in which $Z_i \not\perp Z_j | \boldsymbol{Y}, \boldsymbol{E}$.

# L  IN-DEPTH ANALYSIS ON MORE REALISTIC DATA

## L.1  COLORED MNIST

We compare iCaRL with 1) IRM, 2) two variants of IRMG: F-IRM Game (with $\Phi$ fixed to the identity) and V-IRM Game (with a variable $\Phi$), 3) three variants of ERM: ERM (on entire training data), ERM $e$ (on each environment $e$), and ERM GRAYSCALE (on data with no spurious correlations), and 4) ROBUST MIN MAZ (minimizing the maximum loss across the multiple environments). Table 1 shows that iCaRL outperforms all other baselines on Colored MNIST. However, this dataset seems more difficult because even ERM GRAYSCALE, where the spurious correlation with color is removed, falls well short of the optimum.

## L.2  PACS

We report the results on another one of the widely used realistic datasets for OOD generalization: PACS (Li et al., 2017a). This dataset consists of $9,991$ images of dimension $(3, 224, 224)$ and 7 classes from four domains: art, cartoons, photos, and sketches. We used the exact experimental setting that is described in Gulrajani & Lopez-Paz (2020). We provide results averaged over all possible train and test environment combination for one of the commonly used hyper-parameter tuning procedure: train domain validation. As shown in Table 6, iCaRL achieves state-of-the-art performance when compared to those most popular domain generalization alternatives.

# M  IMPLEMENTATION DETAILS

## M.1  JOINT TRAINING

As described in Section 4.1, we can jointly learn $(\boldsymbol{\theta}, \boldsymbol{\phi})$ by optimizing the following objective:

$$\mathcal{L}_{\text{phase1}}(\boldsymbol{\theta}, \boldsymbol{\phi}) = \mathcal{L}_{\text{phase1}}^{\text{ELBO}}(\boldsymbol{f}, \hat{\boldsymbol{T}}, \hat{\boldsymbol{\lambda}}, \boldsymbol{\phi}) - \mathcal{L}_{\text{phase1}}^{\text{SM}}(\hat{\boldsymbol{f}}, \boldsymbol{T}, \boldsymbol{\lambda}, \hat{\boldsymbol{\phi}}) \tag{62}$$

$$= \mathbb{E}_{p_D}\left[\mathbb{E}_{q_{\boldsymbol{\phi}}(\boldsymbol{Z}|\boldsymbol{X},\boldsymbol{Y},\boldsymbol{E})}\left[\log p_{\boldsymbol{f}}(\boldsymbol{X}|\boldsymbol{Z}) + \log p_{\hat{\boldsymbol{T}},\hat{\boldsymbol{\lambda}}}(\boldsymbol{Z}|\boldsymbol{Y},\boldsymbol{E}) - \log q_{\boldsymbol{\phi}}(\boldsymbol{Z}|\boldsymbol{X},\boldsymbol{Y},\boldsymbol{E})\right]\right]$$

$$- \mathbb{E}_{p_D}\left[\mathbb{E}_{q_{\hat{\boldsymbol{\phi}}}(\boldsymbol{Z}|\boldsymbol{X},\boldsymbol{Y},\boldsymbol{E})}\left[||\nabla_{\boldsymbol{Z}}\log q_{\hat{\boldsymbol{\phi}}}(\boldsymbol{Z}|\boldsymbol{X},\boldsymbol{Y},\boldsymbol{E}) - \nabla_{\boldsymbol{Z}}\log p_{\boldsymbol{T},\boldsymbol{\lambda}}(\boldsymbol{Z}|\boldsymbol{Y},\boldsymbol{E})||^2\right]\right] \tag{63}$$

$$= \mathbb{E}_{p_D}\left[\mathbb{E}_{q_{\boldsymbol{\phi}}(\boldsymbol{Z}|\boldsymbol{X},\boldsymbol{Y},\boldsymbol{E})}\left[\log p_{\boldsymbol{f}}(\boldsymbol{X}|\boldsymbol{Z}) + \log p_{\hat{\boldsymbol{T}},\hat{\boldsymbol{\lambda}}}(\boldsymbol{Z}|\boldsymbol{Y},\boldsymbol{E}) - \log q_{\boldsymbol{\phi}}(\boldsymbol{Z}|\boldsymbol{X},\boldsymbol{Y},\boldsymbol{E})\right]\right]$$

$$- \mathbb{E}_{p_D}\left[\mathbb{E}_{q_{\hat{\boldsymbol{\phi}}}(\boldsymbol{Z}|\boldsymbol{X},\boldsymbol{Y},\boldsymbol{E})}\left[\sum_{j=1}^{n}\left[\frac{\partial^2 p_{\boldsymbol{T},\boldsymbol{\lambda}}(\boldsymbol{Z}|\boldsymbol{Y},\boldsymbol{E})}{\partial Z_j^2} + \frac{1}{2}\left(\frac{\partial p_{\boldsymbol{T},\boldsymbol{\lambda}}(\boldsymbol{Z}|\boldsymbol{Y},\boldsymbol{E})}{\partial Z_j}\right)^2\right]\right]\right]$$

$$+ const. \tag{64}$$

where the last equality is due to the equation in Appendix C.2, and $\hat{\boldsymbol{f}}, \hat{\boldsymbol{T}}, \hat{\boldsymbol{\lambda}}, \hat{\boldsymbol{\phi}}$ are copies of $\boldsymbol{f}, \boldsymbol{T}, \boldsymbol{\lambda}, \boldsymbol{\phi}$ that are treated as constants and whose gradient is not calculated during learning. In practice, $\hat{\boldsymbol{f}}, \hat{\boldsymbol{T}}, \hat{\boldsymbol{\lambda}}, \hat{\boldsymbol{\phi}}$ can be easily implemented through either "`detach`" in PyTorch Paszke et al. (2019) or "`stop_gradient`" in TensorFlow Abadi et al. (2015).

## M.2  THE GENERAL NON-FACTORIZED PRIOR

In the experiments, the general non-factorized prior in Assumption 2 is implemented as follows:

$$p_{\boldsymbol{T},\boldsymbol{\lambda}}(\boldsymbol{Z}|\boldsymbol{Y},\boldsymbol{E}) = \big\langle \underbrace{\text{NN}(\boldsymbol{Z}; \texttt{param1})}_{\boldsymbol{T}_{NN}(\boldsymbol{Z})}, \underbrace{\text{NN}(\boldsymbol{Y},\boldsymbol{E}; \texttt{param2})}_{\boldsymbol{\lambda}_{NN}(\boldsymbol{Y},\boldsymbol{E})}\big\rangle + \big\langle \underbrace{\text{concat}(\boldsymbol{Z},\boldsymbol{Z}^2)}_{\boldsymbol{T}_f(\boldsymbol{Z})}, \underbrace{\text{NN}(\boldsymbol{Y},\boldsymbol{E}; \texttt{param3})}_{\boldsymbol{\lambda}_f(\boldsymbol{Y},\boldsymbol{E})}\big\rangle,$$

where $\langle\cdot,\cdot\rangle$ is the dot product of two vectors, and $\texttt{concat}(\cdot,\cdot)$ means the concatenation of two vectors. Now let us explain each term in details.

Firstly, $\texttt{concat}(\boldsymbol{Z},\boldsymbol{Z}^2)$ is a vector of the latent variables and their squared values, and $\text{NN}(\boldsymbol{Y},\boldsymbol{E}; \texttt{param3})$ is a deep neural network parameterized by $\texttt{param3}$ that computes a vector of natural parameters as a function of $\boldsymbol{Y}$ and $\boldsymbol{E}$. Hence, the term $\big\langle \texttt{concat}(\boldsymbol{Z},\boldsymbol{Z}^2), \text{NN}(\boldsymbol{Y},\boldsymbol{E}; \texttt{param3})\big\rangle$ is equivalent to the factorized exponential family, which also satisfies that each $\boldsymbol{T}_i(Z_i)$ has dimension larger or equal to 2.

Secondly, $\text{NN}(\boldsymbol{Z}; \texttt{param1})$ is a neural network that receives as input a vector of latent variables and outputs another vector representing complicated nonlinear transformations of those variables. $\text{NN}(\boldsymbol{Y},\boldsymbol{E}; \texttt{param2})$ is another neural network that generates a corresponding vector of natural parameters. Hence, the term $\big\langle \text{NN}(\boldsymbol{Z}; \texttt{param1}), \text{NN}(\boldsymbol{Y},\boldsymbol{E}; \texttt{param2})\big\rangle$ will allow this prior to capture the dependencies between the latent variables $\boldsymbol{Z}$.

# N  HYPERPARAMETERS AND ARCHITECTURES

In this section, we describe the hyperparameters and architectures of different models used in different experiments. Unless stated otherwise, we have $\lambda_1 = 1$ and $\lambda_2 = 1$, both of which are selected on training/validation data.

## N.1  SYNTHETIC DATA

We used Adam optimizer for training with learning rate set to 1e-3 and batch size set to 128.

### N.1.1  ERM

**Linear ERM**

- Input layer: Input batch *(batch size, input dimension)*
- Output layer: Fully connected layer, output size = 1

**Nonlinear ERM**

- Input layer: Input batch *(batch size, input dimension)*
- Layer 1: Fully connected layer, output size = 6, activation = ReLU
- Output layer: Fully connected layer, output size = 1

### N.1.2   IRM

**Linear Data Representation $\Phi$**

- Input layer: Input batch *(batch size, input dimension)*
- Output layer: Fully connected layer, output size = 1

**Nonlinear Data Representation $\Phi$**

- Input layer: Input batch *(batch size, input dimension)*
- Layer 1: Fully connected layer, output size = 6, activation = ReLU
- Output layer: Fully connected layer, output size = 1

### N.1.3   F-IRM GAME

**Linear Classifier $w$**

- Input layer: Input batch *(batch size, input dimension)*
- Output layer: Fully connected layer, output size = 1

**Nonlinear Classifier $w$**

- Input layer: Input batch *(batch size, input dimension)*
- Layer 1: Fully connected layer, output size = 6, activation = ReLU
- Output layer: Fully connected layer, output size = 1

### N.1.4   V-IRM GAME

**Linear Data Representation $\Phi$**

- Input layer: Input batch *(batch size, input dimension)*
- Output layer: Fully connected layer, output size = 2

**Nonlinear Data Representation $\Phi$**

- Input layer: Input batch *(batch size, input dimension)*
- Layer 1: Fully connected layer, output size = 6, activation = ReLU
- Output layer: Fully connected layer, output size = 2

**Linear Classifier $w$**

- Input layer: Input batch *(batch size, 2)*
- Output layer: Fully connected layer, output size = 1

**Nonlinear Classifier $w$**

- Input layer: Input batch *(batch size, 2)*
- Layer 1: Fully connected layer, output size = 6, activation = ReLU
- Output layer: Fully connected layer, output size = 1

### N.1.5 ICARL

**NF-iVAE $\lambda_f$-Linear Prior**

- Input layer: Input batch *(batch size, input dimension)*
- Output layer: Fully connected layer, output size = 4

**NF-iVAE $\lambda_f$-Nonlinear Prior**

- Input layer: Input batch *(batch size, input dimension)*
- Layer 1: Fully connected layer, output size = 6, activation = ReLU
- Output layer: Fully connected layer, output size = 4

**NF-iVAE $T_{NN}$-Nonlinear Prior**

- Input layer: Input batch *(batch size, input dimension)*
- Layer 1: Fully connected layer, output size = 6, activation = ReLU
- Output layer: Fully connected layer, output size = 1

**NF-iVAE $\lambda_{NN}$-Nonlinear Prior**

- Input layer: Input batch *(batch size, input dimension)*
- Layer 1: Fully connected layer, output size = 6, activation = ReLU
- Output layer: Fully connected layer, output size = 1

**NF-iVAE Linear Encoder**

- Input layer: Input batch *(batch size, input dimension)*
- Mean Output layer: Fully connected layer, output size = 2
- Log Variance Output layer: Fully connected layer, output size = 2

**NF-iVAE Nonlinear Encoder**

- Input layer: Input batch *(batch size, input dimension)*
- Layer 1: Fully connected layer, output size = 6, activation = ReLU
- Mean Output layer: Fully connected layer, output size = 2
- Log Variance Output layer: Fully connected layer, output size = 2

**NF-iVAE Linear Decoder**

- Input layer: Input batch *(batch size, 2)*
- Mean Output layer: Fully connected layer, output size = output dimension
- Variance Output layer: $0.01 \times \mathbf{1}$, where $\mathbf{1}$ is a vector full of $1$ with the length of output dimension

**NF-iVAE Nonlinear Decoder**

- Input layer: Input batch *(batch size, 2)*
- Layer 1: Fully connected layer, output size = 6, activation = ReLU
- Mean Output layer: Fully connected layer, output size = output dimension
- Variance Output layer: $0.01 \times \mathbf{1}$, where $\mathbf{1}$ is a vector full of $1$ with the length of output dimension

**Linear Classifier $w$**

- Input layer: Input batch *(batch size, 1)*
- Output layer: Fully connected layer, output size = 1

**Nonlinear Classifier $w$**

- Input layer: Input batch *(batch size, 1)*
- Layer 1: Fully connected layer, output size = 6, activation = ReLU
- Output layer: Fully connected layer, output size = 1

## N.2   CMNIST AND CFMNIST

Considering that the results of most baselines come from IRMG (Ahuja et al., 2020a), for a fair comparison, we follow the same setting of IRMG in terms of hyper-parameters and validation considerations. For example, the batch size is set to 256, and the learning rate is $10^{-4}$. We also did not use the test environment data for validation. Please find more details in Ahuja et al. (2020a).

**NF-iVAE $T_{NN}$-Prior**

- Input layer: Input batch *(batch size, input dimension)*
- Layer 1: Fully connected layer, output size = 50, activation = ReLU
- Output layer: Fully connected layer, output size = 45

**NF-iVAE $\lambda_{NN}$-Prior**

- Input layer: Input batch *(batch size, input dimension)*
- Layer 1: Fully connected layer, output size = 50, activation = ReLU
- Output layer: Fully connected layer, output size = 45

**NF-iVAE $\lambda_f$-Prior**

- Input layer: Input batch *(batch size, input dimension)*
- Layer 1: Fully connected layer, output size = 50, activation = ReLU
- Output layer: Fully connected layer, output size = 20

**NF-iVAE $X$-Encoder**

- Input layer: Input batch *(batch size, 2, 28, 28)*
- Layer 1: Convolutional layer, output channels = 32, kernel size = 3, stride = 2, padding = 1, activation = ReLU
- Layer 2: Convolutional layer, output channels = 32, kernel size = 3, stride = 2, padding = 1, activation = ReLU
- Layer 3: Convolutional layer, output channels = 32, kernel size = 3, stride = 2, padding = 1, activation = ReLU
- Output layer: Flatten

**NF-iVAE $(Y, E)$-Encoder**

- Input layer: Input batch *(batch size, input dimension)*
- Output layer: Fully connected layer, output size = 100, activation = ReLU

**NF-iVAE $(X, Y, E)$-Merger/Encoder**

- Input layer: Input batch *(batch size, input dimension)*
- Layer 1: Fully connected layer, output size = 100, activation = ReLU
- Mean Output layer: Fully connected layer, output size = 10
- Log Variance Output layer: Fully connected layer, output size = 10

**NF-iVAE Decoder**

- Input layer: Input batch *(batch size, 10)*
- Layer 1: Fully connected layer, output size = $32 \times 4 \times 4$, activation = ReLU
- Layer 2: Reshape to *(batch size, 32, 4, 4)*
- Layer 3: Deconvolutional layer, output channels = 32, kernel size = 3, stride = 2, padding = 1, outpadding = 0, activation = ReLU
- Layer 4: Deconvolutional layer, output channels = 32, kernel size = 3, stride = 2, padding = 1, outpadding = 1, activation = ReLU
- Layer 5: Deconvolutional layer, output channels = 2, kernel size = 3, stride = 2, padding = 1, outpadding = 1
- Mean Output layer: activation = Sigmoid
- Variance Output layer: $0.01 \times \mathbf{1}$, where $\mathbf{1}$ is a matrix full of 1 with the size of $2 \times 28 \times 28$.

**Classifier $w$**

- Input layer: Input batch *(batch size, 50)*
- Layer 1: Fully connected layer, output size = 100, activation = ReLU
- Output layer: Fully connected layer, output size = 1, activation = Sigmoid

## N.3 VLCS AND PACS

We used the exact experimental setting that is described in Gulrajani & Lopez-Paz (2020). Specifically, we trained our model over all possible train and test environment combination for one of the commonly used hyper-parameter tuning procedure: train domain validation. We use ResNet-50 as an encoder and reverse the architecture of ResNet-50 as a decoder. We set the number of the latent variables to $n = 50$. We do the hyperparameter search by exactly following the guides given in Gulrajani & Lopez-Paz (2020).

