# OpenReview forum: "Invariant Causal Representation Learning for Out-of-Distribution Generalization"
_ICLR.cc/2022/Conference — ICLR 2022 Poster_

### Official Review · Reviewer_Ciwc · 2021-11-01

**Correctness:** 3
**Technical Novelty And Significance:** 3
**Empirical Novelty And Significance:** 3
**Recommendation:** 6
**Confidence:** 3

**Main Review:**

strengths
- a general underlying causal model that covers many real-world scenarios is peoposed
- assumptions of the general causal model are clearly presented and explained
- theoretical guarantee on the identifiability of the latent varibles is provided
- the results of synthetic data experiments are carefully analyzed and can well corroborate the theoretical claims


weakness
- the algorithm seems a bit cumbersome
- there appears to be a technical flaw in Phase 2 of the algorithm
- some popular benchmark data sets are missing

**Summary Of The Paper:**

This paper proposes Invariant Causal Representation Learning (iCaRL), which learns causal representation for OOD generalization and can be viewd as a generalization of IRM to the settings with both nonlinear representations and nonlinear classidiers. The problem is important but part of the algorithm seems not technically correct.

**Summary Of The Review:**

This paper introduces an interesting idea about a general causal model, and presents related assumptions and theoretical results clearly. There seems to be some technical flaws in the proposed algorithm and the experimental results could be improved by adding some popular benchmark  real-world data set.


detailed comments
- the general casual structure (Fig. 1(c)) lasks an edge from E to Y compared with Fig. 1(b). Will this difference affect its expressiveness such that some data-generating processes of Fig. 1(b) can not be characterized by Fig. 1(c)
- when determining Pa(Y) in phase 2, the algorithm compares the p-values of independent testing IndTest(X_i, X_j) and conditional independence testing IndTest(X_i, X_j | Y). Since these two methods derive test statistic from differernt rationale, I was wondering whether the p-values of these two test can be directly compared to conclude the evidence of independence.
- I was wondering whether phase 2 is able to distinguish Y's parents between the its ancestors. it seems an ancestor of a direct cause of Y will also be included in Pa(Y) after phase 2.
- this paper only provedes the results on VLCS data set while existing works usually evaluate the prerformance on at least two real-world data set. It would be much better to include one more real-world data set (e.g. PACS)

---

> ### Author Response · Authors · 2021-11-21
> **To Reviewer Ciwc [1/2]**
>
> Thank you for your comments! Below we provide responses to your questions.
>
> **Q1: the general causal structure (Fig. 1(c)) lacks an edge from E to Y compared with Fig. 1(b). Will this difference affect its expressiveness such that some data-generating processes of Fig. 1(b) can not be characterized by Fig. 1(c)**
>
> A1: In Fig. 1c (which corresponds to our model), we did not draw the edge from E to Y because of Assumption 1d, which is a widely-used default assumption in OOD generalization (Peters et al., 2015; Arjovsky et al., 2019). However, as we explained in the paragraph below Assumption 1, Assumption 1d can be further relaxed to the more practical one that $\mathbb{E}[\boldsymbol{Y} |\boldsymbol{X}_p]$ is invariant across all the environments. That is, given $\boldsymbol{X}_p$, we allow $\boldsymbol{E}$ to only affect the amount of noise in the distribution of $\boldsymbol{Y}$, because that would not change the expected value of $\boldsymbol{Y}$ since the mean of the noise would be zero. In this relaxed version, the general causal structure would allow for an edge from $\boldsymbol{E}$ to $\boldsymbol{Y}$, in which case Fig. 1b is a special case.
>
> **Q2: when determining $\textbf{Pa}(\boldsymbol{Y})$ in phase 2, the algorithm compares the p-values of independent testing $\texttt{IndTest}(X_i, X_j)$ and conditional independence testing $\texttt{IndTest}(X_i, X_j | \boldsymbol{Y})$. Since these two methods derive test statistics from different rationale, I was wondering whether the p-values of these two tests can be directly compared to conclude the evidence of independence.**
>
> A2: Thanks for asking.
>
> Firstly, to clarify, here we emphasize that the dependency between $X_i$ and $X_j$ **increases** after **additionally** conditioning on $\boldsymbol{Y}$, when both of them are causes of $\boldsymbol{Y}$. This is because in the generic case, **additionally** conditioning on $\boldsymbol{Y}$ introduces another spurious correlation between $X_i$ and $X_j$, which will strengthen the dependency between them. In our experiments, this was implemented by comparing the p-values of conditional independence testing $\texttt{IndTest}(X_i, X_j|\boldsymbol{E})$ and conditional independence testing $\texttt{IndTest}(X_i, X_j | \boldsymbol{Y}, \boldsymbol{E})$.
>
> Secondly, comparing the two p-values is a heuristic that worked well in practice (our formulation “it is trivial” may have been unfortunate and has been changed). We believe that using p-values is more sensible than, e.g., using distances in RKHSs (if we use RKHS-based (conditional) independence tests) since p-values live on the same scale and have the same semantics (which would not be the case for the RKHS distances). However, this is still a heuristic, and a more principled approach might try to construct a suitable three-sample test (cf. https://arxiv.org/abs/1511.04581), which would be a whole research project on its own and go beyond the scope of this work.

---

> ### Author Response · Authors · 2021-11-21
> **To Reviewer Ciwc [2/2]**
>
> Thank you for your comments! Below we provide responses to your questions.
>
> **Q3: I was wondering whether phase 2 is able to distinguish $\boldsymbol{Y}$'s parents between the its ancestors. it seems an ancestor of a direct cause of $\boldsymbol{Y}$ will also be included in $\textbf{Pa}(\boldsymbol{Y})$ after phase 2.**
>
> A3: Thank you for pointing this out.
>
> Firstly, as the algorithm is described in the paper, indeed latent variables that are ancestors of a direct cause of Y will be included in $\textbf{Pa}(\boldsymbol{Y})$. However, this does not affect our method's predictive performance since it is based on distinguishing effects from causes of $\boldsymbol{Y}$, and including the ancestors of $\textbf{Pa}(\boldsymbol{Y})$ among the variables used to predict Y will not affect our OOD predictions. To see this, denote by $\textbf{Pa}(\boldsymbol{Y})$ the direct causes of $\boldsymbol{Y}$, and by $\textbf{An}(\boldsymbol{Y})$ the set of latent variables that are its ancestors, **excluding** $\textbf{Pa}(\boldsymbol{Y})$. Under our assumptions about the underlying causal graph (Assumption 1 and Figure 1c), we have that $\boldsymbol{Y} \perp \boldsymbol{E} | \textbf{Pa}(\boldsymbol{Y}), \textbf{An}(\boldsymbol{Y})$, where $\perp$ denotes probabilistic independence and which implies that $p(\boldsymbol{Y}|\textbf{Pa}(\boldsymbol{Y}), \textbf{An}(\boldsymbol{Y}))$ is still invariant across all the environments. This means that additionally including $\boldsymbol{An}(\boldsymbol{Y})$ as inputs to a predictor for Y will not affect the final OOD prediction. Thank you for pointing this out, and we have clarified this in the updated version of the paper.
>
> In addition, we do have a simple heuristic to remove $\textbf{An}(\boldsymbol{Y})$ in Phase 2, in cases where this is desired. We first conduct a constraint-based causal discovery method to learn a Markov equivalence class of DAGs. This gives us the direct neighbors of $\boldsymbol{Y}$ in the causal graph ($\boldsymbol{Y}$'s parents, without any other additional ancestors, and $\boldsymbol{Y}$'s children). That is, among $\boldsymbol{Y}$’s neighbors there are no elements belonging to $\textbf{An}(\boldsymbol{Y})$, since they have been removed by the constraint-based method. We can then further remove $\boldsymbol{Y}$'s children from this set of nodes by using our above approach of comparing p-values between different pairs of latent variables (only $\boldsymbol{Y}$'s parents will have an increment in dependence when additionally conditioning on $\boldsymbol{Y}$). Any nodes returned by our method that are not returned by this heuristic method are potential ancestors of a direct cause of $\boldsymbol{Y}$.
>
> We have compared the heuristic method from the previous paragraph with the method currently described in our paper, for the experiments in the paper. Both methods identified the same set of the direct causes of $\boldsymbol{Y}$, i.e., our method did not identify ancestors of a direct cause of $\boldsymbol{Y}$ as causes of $\boldsymbol{Y}$. We acknowledge that this can be due to the choice of our experiments.
>
> We have clarified this point and included the heuristic method above as a way to eliminate any potential ancestors of direct causes of $\boldsymbol{Y}$ in Phase 2.
>
> **Q4: this paper only provides the results on VLCS data set while existing works usually evaluate the performance on at least two real-world data sets. It would be much better to include one more real-world data set (e.g. PACS)**
>
> A4: We have run additional experiments on PACS. We used the exact experimental setting that is described in Gulrajani & Lopez-Paz (2020). We provide results averaged over all possible train and test environment combinations for one of the commonly used hyper-parameter tuning procedure: train domain validation. As shown in the table below, iCaRL achieves state-of-the-art performance when compared to those most popular domain generalization alternatives. We also added the results on PACS to the appendix.
>
> | METHOD | TEST  |
> | --- | --- |
> | ERM | 85.7 ± 0.5 |
> | IRM | 84.4 ± 1.1 |
> | DRO (Sagawa et al., 2019) | 84.1 ± 0.4 |
> | Mixup (Yan et al., 2020) | 84.3 ± 0.5 |
> | CORAL (Sun & Saenko, 2016)  | 86.0 ± 0.2 |
> | MMD (Li et al., 2018b)  | 85.0 ± 0.2 |
> | DANN (Ganin et al., 2016)  | 84.6 ± 1.1 |
> | C-DANN (Li et al., 2018c) | 82.8 ± 1.5 |
> | LaCIM (Sun et al., 2020) | 83.5 ± 1.2 |
> | **iCaRL (ours)** | **88.7 ± 0.6** |

---

### Official Review · Reviewer_rp1x · 2021-11-03

**Correctness:** 3
**Technical Novelty And Significance:** 3
**Empirical Novelty And Significance:** 3
**Recommendation:** 6
**Confidence:** 3

**Main Review:**

Section 4.1 presents the theoretical foundation, with a complete and clear proof. This is a valuable contribution.

Section 4.2 is less than rigorous: a) “one observation is that for any two latent variables Xi and Xj , only when both are causes of Y do we have that the dependency between them increases after conditioning on Y .” Note that Xi and Xj are actually correlated due to the confounder E, how to justify this claim? Or do you mean conditional on E? b) ”when there exist at least
two causal latent variables, it is trivial to test all pairs of latent variables with independence testing (Gretton et al., 2007) and conditional independence testing (Zhang et al., 2012) to discover all of them by comparing p-values from these two tests.” I am not sure whether the p-values can be compared. Also, p-value itself is not a statistical measure. Please justify the methods in this section.

Section 4.3 gives the algorithm wrt. both training datasets and new test data. Here is my main concern: I wonder whether this method is for DG and OOD generalization, or indeed a DA method. Note that “ Phase 3:  … When in a new environment, we first infer Pa(Y) from O by solving Eq. (12) and then leverage the learned w for prediction.” This means, the method requires knowledge of test domain data, although in a not very complicated way. To me, if test domain data are needed in adjusting the algorithm, then it should be a DA method. Note that this is different from simple normalization, as this operation can be included in the trained model. Further, what if we have a test dataset as a mixture of several domains, then is this method affected? Please clarify.

Minor concerns and questions:
1.	The causal graph is explained well wrt practical considerations. However, it is not clear how practical the assumptions in Theorem 1 and 2 are. For instance, is it true to have O=f(x) injective in VLCS datasets? Why is it needed to assume n <= d?
2.	“It also makes sense in Assumption 1c that the generative mechanism p(O| X) is invariant across all the environments. Otherwise, it is impossible to infer X from O in any unseen environment” Is it possible to assume only $P(O|X_pa)$ remains invariant?
3.	Footnote 7 is missing. Also maybe move some footnotes to the main text in later version (e.g., camera ready version if accepted)? Footnote kind of affecs reading flow.
4.	In practical algorithms, how to select the dimension of X?
5.	(very minor concern) maybe to include one or two more real datasets like PACS?


**Summary Of The Paper:**

This paper proposes invariant Causal Representation Learning (iCaRL) for OOD generalization in the nonlinear setting. The work extends iVAE to a somewhat more general setting and shows the direct cause of the target can be discovered. iCARL is then developed based on the direct cause. . Extensive experiments verify the effectiveness of the proposed method.

**Summary Of The Review:**

In summary, the paper has adequate theoretical contributions in terms of identifiability. The algorithm part seems less justified. Also, the method appears to be a DA method, not for OOD generalization, thus authors should compare their algorithm to recent DA methods. I look forward to authors' response.

== after reading response ==
Based on the current response, I decide to increase my evaluation.

---

> ### Author Response · Authors · 2021-11-21
> **To Reviewer rp1x [1/3]**
>
> Thank you for your comments! Below we provide responses to your questions.
>
> **Q1: a) “one observation is that for any two latent variables $X_i$ and $X_j$ , only when both are causes of $\boldsymbol{Y}$ do we have that the dependency between them increases after conditioning on $\boldsymbol{Y}$ .” Note that $X_i$ and $X_j$ are actually correlated due to the confounder $\boldsymbol{E}$, how to justify this claim? Or do you mean conditional on $\boldsymbol{E}$?**
>
> A1: To clarify, here we emphasize that the dependency between $X_i$ and $X_j$ **increases** after **additionally** conditioning on $\boldsymbol{Y}$, when both of them are causes of $\boldsymbol{Y}$. This is because in the generic case, **additionally** conditioning on $\boldsymbol{Y}$ introduces another spurious correlation between $X_i$ and $X_j$, which will strengthen the dependency between them. This can be implemented by comparing the p-values of conditional independence testing $\texttt{IndTest}(X_i, X_j|\boldsymbol{E})$ and conditional independence testing $\texttt{IndTest}(X_i, X_j | \boldsymbol{Y}, \boldsymbol{E})$.
>
> **Q2: b) ”when there exist at least two causal latent variables, it is trivial to test all pairs of latent variables with independence testing (Gretton et al., 2007) and conditional independence testing (Zhang et al., 2012) to discover all of them by comparing p-values from these two tests.” I am not sure whether the p-values can be compared. Also, p-value itself is not a statistical measure. Please justify the methods in this section.**
>
> A2: Thanks for asking. Comparing the two p-values is a heuristic that worked well in practice (our formulation “it is trivial” may have been unfortunate and has been changed). We believe that using p-values is more sensible than, e.g., using distances in RKHSs (if we use RKHS-based (conditional) independence tests) since p-values live on the same scale and have the same semantics (which would not be the case for the RKHS distances). However, this is still a heuristic, and a more principled approach might try to construct a suitable three-sample test (cf. https://arxiv.org/abs/1511.04581), which would be a whole research project on its own and go beyond the scope of this work.

---

> > ### Comment · Reviewer_rp1x · 2021-11-21
> > **An additional comment on phase 2**
> >
> > Thanks for response, but this does not fully solve my concern on phase 2.
> >
> > I do not think picking a theoretically sound approach in phase 2 would be beyond the scope of the paper. As previously stated, it may not be statistically correct to compare the p-values of two different CI tests, and p-value is not a CI measure or metric. That is, even $CI(X_i, X_j|E)< CI (X_i, X_j|Y,E)$ is true, the followed approach might fail, even with infinite data.
> >
> > “We believe that using p-values is more sensible than, e.g., using distances in RKHSs (if we use RKHS-based (conditional) independence tests) since p-values live on the same scale and have the same semantics (which would not be the case for the RKHS distances).”
> >
> > If the RKHS based distance is indeed a CI metric, that this should not be a problem.

---

> > > ### Author Response · Authors · 2021-11-23
> > > **Response to the additional comment**
> > >
> > > Thanks for your comments!
> > >
> > > **Regarding discovering the parents of $\boldsymbol{Y}$ in Phase 2.**
> > >
> > > Phase 2 can be thought of as a standard causal discovery task over $\boldsymbol{X}$ and $\boldsymbol{Y}$, since we allow for arbitrary connections between them as long as the resulting graph is a DAG. In the general case, there are no theoretical guarantees that existing causal discovery algorithms, be it constraint-based approaches (e.g., PC [1], FCI [1]) or score-based approaches (e.g., NOTEARS [2,3], GraN-DAG [4], RL-BIC [5] ), are able to fully discover all the direct causes of a given target (e.g., $\boldsymbol{Y}$ in our case). Precisely, constraint-based approaches only learn a Markov equivalence class of DAGs from data, and score-based approaches are not even guaranteed to learn a correct DAG in most cases. Therefore, how to discover the causes of $\boldsymbol{Y}$ with theoretical guarantees is one of the fundamental challenges and still an open question in the causal discovery area. It indeed goes beyond the scope of our paper. As such, we proposed a heuristic which actually worked quite well in our experiments.
> > >
> > > In fact, in our experiments, we verified this point by comparing our heuristic approach with the PC algorithm (one of the widely used causal discovery algorithms) in the task of discovering the parents of $\boldsymbol{Y}$. We found that the PC algorithm can only discover a subset of $\boldsymbol{Y}$’s direct causes, whilst our heuristic approach can discover all of them in our experiments. Note that, on the image data, we can assess whether or not each $X_i$ is a cause of $\boldsymbol{Y}$ by visualizing the generated images through performing intervention upon it. This demonstrates that the heuristic approach of comparing p-values works better in our experiments.
> > >
> > > We do believe that this phase in our method would benefit a lot from the development of the whole causal discovery area.
> > >
> > > **Regarding RKHS-based distance vs p-values.**
> > >
> > > The RKHS-based distance for independence testing is used to look at distances with respect to the distribution given by the product of the marginals (independent distribution). Therefore, for conditional distributions, intuitively, we would have that $\texttt{IndTest}(X_i,X_j|\boldsymbol{E})$ looks at the distance between $p(X_i,X_j|\boldsymbol{E})$ and $p(X_i|\boldsymbol{E})p(X_j|\boldsymbol{E})$, while $\texttt{IndTest}(X_i,X_j|\boldsymbol{Y},\boldsymbol{E})$ looks at the distance between $p(X_i,X_j|\boldsymbol{Y},\boldsymbol{E})$ and $p(X_i|\boldsymbol{Y},\boldsymbol{E})p(X_j|\boldsymbol{Y},\boldsymbol{E})$, each of which is **in their own kernel space** where **the distributions are respectively embedded**. In this sense, the RKHS-based distances are less meaningful on their own, since "a difference of 5 (or whatever)" is not so interpretable (Is "5" big? Small?). By contrast, the p-value is "how unlikely is the statistic under the null hypothesis," so it gives a ranking of the RKHS-based distances (e.g., MMDs/HSICs) that is meaningful. Hence, a p-value going from 0.01 to 0.001 would indicate an important difference in distinguishing from the null hypothesis. That is why we believe that using p-values is more sensible than, e.g., using distances in RKHSs (if we use RKHS-based (conditional) independence tests) since p-values live on the same scale and have the same semantics (which would not be the case for the RKHS distances).
> > >
> > > **References:**
> > >
> > > [1] Spirtes, P., Glymour, C. N., Scheines, R., and Heckerman, D. (2000). Causation, prediction, and search. MIT press.
> > >
> > > [2] Zheng, X., Aragam, B., Ravikumar, P., and Xing, E. P. (2018). DAGs with NO TEARS: Continuous Optimization for Structure Learning. In Advances in Neural Information Processing Systems.
> > >
> > > [3] Zheng, X., Dan, C., Aragam, B., Ravikumar, P., and Xing, E. P. (2020). Learning Sparse Nonparametric DAGs. International Conference on Artificial Intelligence and Statistics.
> > >
> > > [4] Lachapelle, S., Brouillard, P., Deleu, T., and Lacoste-Julien, S. (2019). Gradient-based neural dag learning. arXiv preprint arXiv:1906.02226.
> > >
> > > [5] Zhu, S., Ng, I., and Chen, Z. (2019). Causal discovery with reinforcement learning. arXiv preprint arXiv:1906.04477.

---

> > > > ### Comment · Reviewer_rp1x · 2021-11-23
> > > > **Clarifications**
> > > >
> > > > First, thanks for the timely response. Sorry for bringing misunderstanding. I mean I feel OK with the use of PC. My concern is whether it is statistically correct to compare the p-values of two different CI test statisics.
> > > >
> > > > BTW, there is normalized RKHS based distance so that they live on the same scale.

---

> > > > > ### Author Response · Authors · 2021-11-24
> > > > > **Response to clarifications**
> > > > >
> > > > > Thanks for the clarification and the suggestion! We did look for such methods, and reached out to experts in the field. It is a nontrivial problem to perform such a rescaling so that it applies to both conditional kernel mean embeddings for the two different CI test statistics and leads to numbers that are justifiably comparable. Comparing p-values can be thought of as one such method, but we would be happy to receive pointers to better solutions.

---

> ### Author Response · Authors · 2021-11-21
> **To Reviewer rp1x [2/3]**
>
> Thank you for your comments! Below we provide responses to your questions.
>
> **Q3: Section 4.3 gives the algorithm wrt. both training datasets and new test data. Here is my main concern: I wonder whether this method is for DG and OOD generalization, or indeed a DA method. Note that “ Phase 3: ... When in a new environment, we first infer $\textbf{Pa}(\boldsymbol{Y})$ from $\boldsymbol{O}$ by solving Eq. (12) and then leverage the learned $\boldsymbol{w}$ for prediction.” This means, the method requires knowledge of test domain data, although in a not very complicated way. To me, if test domain data are needed in adjusting the algorithm, then it should be a DA method. Note that this is different from simple normalization, as this operation can be included in the trained model. Further, what if we have a test dataset as a mixture of several domains, then is this method affected? Please clarify.**
>
> A3: Thank you for asking. Let us first clarify why our method is for DG rather than for DA.
>
> In our iCaRL framework, the model parameters are $(\boldsymbol{\theta}=(\boldsymbol{f}, \boldsymbol{T}, \boldsymbol{\lambda}), \boldsymbol{\phi}, \boldsymbol{w})$, where $\boldsymbol{f}$ are the parameters of the decoder $p_{\boldsymbol{f}}(\boldsymbol{O}|\boldsymbol{X})$, $\boldsymbol{T}$ and $\boldsymbol{\lambda}$ the parameters of the prior $p_{\boldsymbol{T}, \boldsymbol{\lambda}}(\boldsymbol{X}|\boldsymbol{Y}, \boldsymbol{E})$, $\boldsymbol{\phi}$ the parameters of the encoder $q_{\boldsymbol{\phi}}(\boldsymbol{X}|\boldsymbol{O}, \boldsymbol{Y}, \boldsymbol{E})$, and $\boldsymbol{w}$ the parameters of the invariant classifier. All these parameters are learned only from training domain data, without the need of test domain data. More specifically, the parameters $(\boldsymbol{\theta}=(\boldsymbol{f}, \boldsymbol{T}, \boldsymbol{\lambda}), \boldsymbol{\phi})$ are learned from training domain data by solving Eq. (10), and the parameters $\boldsymbol{w}$ are learned also from training domain data by solving Eq. (11).
>
> When in a new environment, we do not need to learn any of the model parameters $(\boldsymbol{\theta}=(\boldsymbol{f}, \boldsymbol{T}, \boldsymbol{\lambda}), \boldsymbol{\phi}, \boldsymbol{w})$, and the only thing we need to do is just to infer $\textbf{Pa}(\boldsymbol{Y})$ (i.e., $\boldsymbol{X}_p$) from $\boldsymbol{O}$ by solving Eq. (12). Note that, while solving Eq. (12), the model parameter $\boldsymbol{f}$ learned from training domain data is fixed, and the hyperparameters $(\lambda_1, \lambda_2)$ were selected on validation data. We only treat $\boldsymbol{X}_p$ and $\boldsymbol{X}_c$ as latent variables to be optimized from test domain data. Hence, solving Eq. (12) is an inference step, rather than a learning step. Once we infer $\boldsymbol{X}_p$ from $\boldsymbol{O}$ in the new environment, we can directly leverage the learned $\boldsymbol{w}$ for prediction. Apparently, during the whole procedure, the test domain data is only used to infer $\boldsymbol{X}_p$ from $\boldsymbol{O}$ by solving Eq. (12) in the new environment. Therefore, our method is for DG rather than for DA.
>
> When we have a test dataset as a mixture of several domains, our method will not be affected. This is because the domain index $\boldsymbol{E}$ is not required to solve Eq. (12) in the test dataset.
>
> Hopefully we have addressed your main concern. Let us know if you have more questions regarding this point.

---

> > ### Comment · Reviewer_rp1x · 2021-11-21
> > **An additional question regarding whether it's DA or DG setting**
> >
> > Thanks for the respsonse. I understand that the model parameters are fixed, after solving Eqs. (10) and (11) from training data. I have an additional question regarding whether it's DA or DG
> >
> > Let me clarify the DG or DA setting. In my opinion, a difference lies in the availability of the test domain. If a number of test data (so that the distribution of the test domain data can be well represented) are provided or needed, then it is a DA setting.
> >
> > In phase 3 when solving (12), how accurate would it be if only **one test data sample** is given? Or whether a number of samples are required so that the inference of $pa$ can be statistically accurate? In IRM, we can pass the test data one by one, or pass all the test data, for inference, and the result keeps the same. It seems to me that the inference results of the whole algorithm of iCaRL may be different.
> >
> >
> > BTW, "the hyperparameters \labmda_1, labmda_2 were selected on validation data". I assumed you mean the two parameters are also obtained based on the training data? This statement seems not mentioned in the main text.

---

> > > ### Author Response · Authors · 2021-11-23
> > > **Response to the additional question**
> > >
> > > Thanks for asking!
> > >
> > > Yes, we also agree that the difference between DG and DA lies in the availability of the test domain. Since both our model parameters and hyperparameters (including $\lambda_1$ and $\lambda_2$, clarified in the paper) are obtained on the training/validation data, **without the need of test data**, that is why we believe that our method is for DG, rather than DA. The test data are only provided to infer $\boldsymbol{X}_p$ from $\boldsymbol{O}$ during the test time.
> > >
> > > In Phase 3, during the test time, Eq. (12) is solved independently for each data point in the test data, to infer $\boldsymbol{X}_p$ for that data point. The fact that we have just one or a number of test data points is irrelevant. In other words, we can either pass the test data one by one or pass all the test data for inference, and the results are the same.
> > >
> > > We hope this answered your question. Let us know if you have more questions regarding this point.

---

> > > > ### Comment · Reviewer_rp1x · 2021-11-23
> > > > **Thanks. Still curious about the details**
> > > >
> > > > Thanks again for the timely response, which mostly solves my major concern. Before increasing my score, let me ask a further question.
> > > >
> > > > "In Phase 3, during the test time, Eq. (12) is solved independently for each data point in the test data, to infer
> > > >  for that data point. The fact that we have just one or a number of test data points is irrelevant. In other words, we can either pass the test data one by one or pass all the test data for inference, and the results are the same".
> > > >
> > > > Please mention this in the paper. And, I am curious about whether they could be the same. Let's think about a datatum from training data of domain A, but it lies closer to domain B. Or maybe domain A and domain B has some overlap and the datum is from the intersection. Now, as in phase 2, the inferred $pa(Y^e)$ may be different. Then using the single datatum could lead to $pa(Y)|e=B$, but when using all the data of domain A in (12) we may have $pa(Y)|e=A$ for that datatum.
> > > >
> > > > ps: As in the notation with the supscript $e$, I suppose that $pa(Y^e)$ is different for different $e$. Correct me I get wrong here. If $pa(Y)$ does not rely on $e$, then what is the multi-domain setting useful here? It seems that you can only use single domain data for training?

---

> > > > > ### Author Response · Authors · 2021-11-24
> > > > > **Response to the further question**
> > > > >
> > > > > Thanks for asking! We will mention that in the final version. Let us clarify why this is the case. We are assuming that the function $\boldsymbol{f}$ mapping $\boldsymbol{X}$ to $\boldsymbol{O}$ is assumed to be **injective** and, therefore, we should in principle be able to estimate $\boldsymbol{X}$ from $\boldsymbol{O}$ only **without access to $\boldsymbol{Y}$ or $\boldsymbol{E}$**. Note that our estimate of $\boldsymbol{f}$ in our learned neural network models is also very likely to be injective because it maps a low-dimensional $\boldsymbol{X}$ to a high-dimensional $\boldsymbol{O}$ and the chance that two points in $\boldsymbol{X}$ space map into the same point in $\boldsymbol{O}$ space is very low. **The injectivity of $\boldsymbol{f}$ explains why having access to more data points at test time would not help, which has also been verified in our experiments.** The question is then why are $\boldsymbol{Y}$ and $\boldsymbol{E}$ necessary at all **at training time** to infer $\boldsymbol{X}$ when learning our latent variable model? The answer is to have identifiability guarantees on the latent variables $\boldsymbol{X}$. That is, by conditioning to $\boldsymbol{Y}$ and $\boldsymbol{E}$ in both the prior and in the recognition network, we can converge to the right form for the function $\boldsymbol{f}$ (one that guarantees that the recovered latent variables are equal to the original ones up to a simple transformation).
> > > > >
> > > > > We hope this answered your question. Let us know if you have more questions regarding this point.

---

> ### Author Response · Authors · 2021-11-21
> **To Reviewer rp1x [3/3]**
>
> Thank you for your comments! Below we provide responses to your questions.
>
> ### **Regarding Minor concerns and questions**
>
> **Q4: The causal graph is explained well wrt practical considerations. However, it is not clear how practical the assumptions in Theorem 1 and 2 are. For instance, is it true to have $\boldsymbol{O}=\boldsymbol{f}(\boldsymbol{X})$ injective in VLCS datasets? Why is it needed to assume $n \leq d$?**
>
> A4: Thank you for asking. Since $\boldsymbol{f}$ maps a low dimensional latent space $\boldsymbol{X}$ into a high-dimensional observed space $\boldsymbol{O}$ (this is also why $n \leq d$ is assumed here), it is quite likely that $\boldsymbol{f}$ will be injective or very close to injective in practice because the probability that two points in the low dimensional latent space are mapped by $\boldsymbol{f}$ to the same point in the high-dimensional observed space is going to be very low. The good results obtained by our method in various experiments seem to corroborate this.
>
> **Q5: “It also makes sense in Assumption 1c that the generative mechanism $p(\boldsymbol{O}| \boldsymbol{X})$ is invariant across all the environments. Otherwise, it is impossible to infer $\boldsymbol{X}$ from $\boldsymbol{O}$ in any unseen environment” Is it possible to assume only $p(\boldsymbol{O}|\boldsymbol{X}_p)$ remains invariant?**
>
> A5: From Figure 1c, we can read that when $\boldsymbol{X}_c$ depends on $\boldsymbol{E}$ (i.e., there exist edges from $\boldsymbol{E}$ to $\boldsymbol{X}_c$), $\boldsymbol{O}$ is not independent of $\boldsymbol{E}$ given $\boldsymbol{X}_p$ alone, and thus $p(\boldsymbol{O}|\boldsymbol{X}_p)$ will not remain invariant across environments. Conversely, if we assume that $p(\boldsymbol{O}|\boldsymbol{X}_p)$ remains invariant, it means that $\boldsymbol{X}_c$ does not depend on $\boldsymbol{E}$, which would be quite limited in the real world applications.
>
> **Q6: Footnote 7 is missing. Also maybe move some footnotes to the main text in later version (e.g., camera ready version if accepted)? Footnote kind of affects reading flow.**
>
> A6: Thanks for the suggestion. We will move them to the main text in the final version. Btw, footnote 7 is “The highly unlikely single-cause case is left in Appendix G”.
>
> **Q7: In practical algorithms, how to select the dimension of X?**
>
> A7: We treat the dimension of $\boldsymbol{X}$ as a hyperparameter and use the ELBO on validation data as a proxy to select it.
>
> **Q8: (very minor concern) maybe to include one or two more real datasets like PACS?**
>
> A8: We have run additional experiments on PACS. We used the exact experimental setting that is described in Gulrajani & Lopez-Paz (2020). We provide results averaged over all possible train and test environment combinations for one of the commonly used hyper-parameter tuning procedures: train domain validation. As shown in the table below, iCaRL achieves state-of-the-art performance when compared to those most popular domain generalization alternatives. We also added the results on PACS to the appendix.
>
>
>
> | METHOD | TEST  |
> | --- | --- |
> | ERM | 85.7 ± 0.5 |
> | IRM | 84.4 ± 1.1 |
> | DRO (Sagawa et al., 2019) | 84.1 ± 0.4 |
> | Mixup (Yan et al., 2020) | 84.3 ± 0.5 |
> | CORAL (Sun & Saenko, 2016)  | 86.0 ± 0.2 |
> | MMD (Li et al., 2018b)  | 85.0 ± 0.2 |
> | DANN (Ganin et al., 2016)  | 84.6 ± 1.1 |
> | C-DANN (Li et al., 2018c) | 82.8 ± 1.5 |
> | LaCIM (Sun et al., 2020) | 83.5 ± 1.2 |
> | **iCaRL (ours)** | **88.7 ± 0.6** |

---

> > ### Public Comment · ~Bin_Deng1 · 2022-07-14
> > **How to insure the the funtion O=f(X) to be injective?**
> >
> > Thanks for the excellent work, and I believe it would be a great step for addressing the nonlinear OOD generalization problem. Here I am wonder how practical would be to make the function $O=f(X)$ injective? I have two concerns:
> >
> > 1. In theory, we have to make $f$ to be injective to ensure the identifiablity of the model. However, when using VAE-based method for likehood maximization, as like the Eq. (8), we maximize $\mathbb{E}_{q_\phi (X|O,Y,E)}\log p_f(O|X)$, and this means that for all support points of distribution $q_\phi (X|O,Y,E)$, we want $f(X)$ to be the same value of $O$, which means the $f$ is being optimized to be non-injective? Whether this process would contradict the theory in the practical use?
> >
> > 2. Since $f$ is usually optimized as a neural network that is highly non-convex, how to insure $f$ to be injective in practice?

---

> ### Author Response · Authors · 2021-11-24
> **Thank you.**
>
> Thank you for raising your score. We will incorporate the suggested changes in the manuscript.

---

### Official Review · Reviewer_it5B · 2021-11-08

**Correctness:** 4
**Technical Novelty And Significance:** 4
**Empirical Novelty And Significance:** 4
**Recommendation:** 8
**Confidence:** 3

**Main Review:**

**Strengths**

This paper has significant technical and empirical novelty. Moreover, it is very well written and communicates the central contributions clearly.

**Weaknesses**

Overall, I only have 2 issues with this paper. **First**, a very minor concern regarding readability. Since your solution builds off of the VAE literature in part, I think that it would be more readable to use $X$ for observed random variables and $Z$ for latent random variables, rather than $O$ and $X$ respectively. While $O$ makes sense since it is the first letter in observed, I think it is burdensome to ask computer science readers to disassociate it from its usage in "Big O notation".

**Second**, and more significantly. I'm not convinced that "Out-of-Distribution Generalization" is a sensible term, and I'm also concerned that it is misleading. I think you clearly present one graphical model that defines one distribution. All environments are a part of that distribution, and I would argue that unseen environments are out-of-sample rather than out-of-distribution.

For me, out-of-distribution would correspond to a fundamental change in the relationship between variables. For example, let's say $Y$ is a binary variable indicating whether or not a sequence of characters $O$ signifies addition. Under one distribution the plus sign (+) (represented by some $X_p$) would have a strong causal association with $Y$. This would be an invariant causal association across whether the characters were printed, hand written, displayed on a screen, blue, purple, etc. If we never observed an orange "+" during training, under the assumption of the same distribution (graphical model) it makes sense that we would want to generalize to this *out-of-sample* environment. But what if we see samples from a different graphical model, say one in which $Y$ now signifies the logical or operation and under this distribution all plus signs are instead associated with that $Y$? If our original model saw samples from this distribution it would confidently and incorrectly classify such examples as signifying addition, thus failing to generalize to the new distribution. This is just one example of why I think it is incorrect to characterize what this method does as out-of-distribution generalization. I am open to further discussion about this point, but I think it would be more precise to use the term out-of-sample in place of out-of-distribution.

**Summary Of The Paper:**

This paper proposes a general framework for so-called "out-of-distribution generalization" that can handle non-linear associations and causal links between variables. It provides proofs for identifiability and generalization. It proposes a general causal model that is sensible for many prediction problems. It proposes a conditionally non-factorized prior. It provides a tractable method that can be used in practice.

**Summary Of The Review:**

This paper has significant technical and empirical novelty. Therefore I recommend acceptance. However, I think we need to have a discussion about whether Out-of-Distribution Generalization is a sensible term.

---

> ### Author Response · Authors · 2021-11-21
> **To Reviewer it5B**
>
> Thank you for your comments! Below we provide responses to your questions.
>
> **Q1: First, a very minor concern regarding readability. Since your solution builds off of the VAE literature in part, I think that it would be more readable to use $\boldsymbol{X}$ for observed random variables and $\boldsymbol{Z}$ for latent random variables, rather than $\boldsymbol{O}$ and $\boldsymbol{X}$ respectively. While $\boldsymbol{O}$ makes sense since it is the first letter in observed, I think it is burdensome to ask computer science readers to disassociate it from its usage in "Big O notation".**
>
> A1: Thank you for the suggestion. In the final submission, we will replace $\boldsymbol{O}$ with $\boldsymbol{X}$ for observed random variables and $\boldsymbol{X}$ with $\boldsymbol{Z}$ for latent random variables. At this moment, since all the reviews have already referred to $\boldsymbol{O}$ and $\boldsymbol{X}$ for observed and unobserved random variables respectively, we still keep using the current notations during the discussion to avoid any notational misunderstanding.
>
> **Q2: Second, regarding the term "Out-of-Distribution Generalization"**
>
> A2: Thank you for asking. Let us clarify more regarding the term “out-of-distribution”.
>
> First, to clarify, in our causal graph shown in Figure 1c, dashed lines denote the edges which might vary across environments and even be absent in some scenarios, as long as Assumption 1 is satisfied. For convenience, let us recap Assumption 1 here, where $\perp$ denotes probabilistic independence:
>
> >Assumption 1. (a) $X_i$ depends on one or both of $\boldsymbol{Y}$ and $\boldsymbol{E}$ for any i; (b) The causal graph containing $\boldsymbol{X}$ and $\boldsymbol{Y}$ is a DAG; (c) $\boldsymbol{O} \perp \boldsymbol{Y}, \boldsymbol{E}|\boldsymbol{X}$, implying that $p(\boldsymbol{O}|\boldsymbol{X})$ is invariant across all the environments; (d) $\boldsymbol{Y} \perp \boldsymbol{E} | \boldsymbol{X}_p$, implying that $p(\boldsymbol{Y}|\boldsymbol{X}_p)$ is invariant across all the environments.
>
> In other words, under Assumption 1, for those dashed edges we allow them to exist in one environment but to change or even vanish in another environment, which exactly corresponds to a fundamental change in the relationship between variables, as you mentioned. Technically, in an environment $e \in \boldsymbol{E}$, for each $X_i$ its corresponding causal module $p^e(X_i|\textbf{Pa}(X_i)) \doteq p(X_i|\textbf{Pa}(X_i), \boldsymbol{E}=e)$ can be represented by the following structural causal model:
> $$
> X_i = f_i(\textbf{Pa}^e(X_i), \epsilon^e_i; \boldsymbol{\theta}^e_i),
> $$
>
> where $\epsilon$ is a disturbance term and has a non-zero variance (i.e., the model is not deterministic), and $\boldsymbol{\theta}$ denotes the effective parameters in the model/mechanism $f$. That they all have the superscript $e$ explicitly indicates that all of them could be affected by $e$ (i.e., they might vary or even vanish across environments). Any change on these three terms in the new environment $e^*$ will produce a different distribution $p^{e^*}(X_i|\textbf{Pa}(X_i))$, meaning that the relationship between $X_i$ and $\textbf{Pa}(X_i)$ changes in the new environment $e^*$. It is worth noting that we also allow $\textbf{Pa}^{e^*}(X_i)$ to be an empty set and then $p^{e^*}(X_i|\textbf{Pa}(X_i))$ reduces to the marginal distribution $p^{e^*}(X_i)$, in which case the edges between $X_i$ and $\textbf{Pa}(X_i)$ vanish in the new environment $e^*$. While the environment can change the distribution of the causal parents of $\boldsymbol{Y}$, the causal mechanism (i.e., $p(\boldsymbol{Y}|\boldsymbol{X}_p) \doteq p(\boldsymbol{Y}|\textbf{Pa}(\boldsymbol{Y}))$) that generates $\boldsymbol{Y}$ from its causal parents remains constant across environments.
>
> Second, in machine learning, the term “out-of-distribution” (OOD) indicating that samples come from a different joint distribution more often contrasts with the term “independent and identically distributed” (IID) indicating that samples come from the same joint distribution. Through the analysis above, we know that any change on the three terms $(\textbf{Pa}(X_i), \epsilon, \boldsymbol{\theta})$ for any $X_i$ in the new environment $e^*$ will produce a different distribution $p^{e^*}(X_i|\textbf{Pa}(X_i))$. As a result, we will have a different joint distribution $p^{e^*}(\boldsymbol{O}, \boldsymbol{Y})$. That’s why we use the term “out-of-distribution”.
>
> **Back to your simple example “But what if we see samples from a different graphical model, say one in which Y now signifies the logical or operation”, in this example we think that Assumption 1d is violated, that is, the causal mechanism $p(\boldsymbol{Y}|\boldsymbol{X}_p)$ is no longer invariant.**
>
> Hopefully we have cleared up your doubts. Let us know if you have more questions regarding this point. We are also open to further discussion.

---

### Author Response · Authors · 2021-11-21
**To All Reviewers**

Thanks for the encouraging reviews, stating that “the problem is important”, “this paper has significant technical and empirical novelty”, is “very well written and communicates the central contributions clearly”, and “extensive experiments verify the effectiveness of the proposed method.” Below, we include our point-by-point responses. We also include experimental evidence for an additional real-world problem.

We are happy to answer any further questions.

---

### Decision · Program_Chairs · 2022-01-20

**Decision:**

Accept (Poster)

**Comment:**

The paper sets up a complex algorithm for out-of-distribution generalization. The algorithm requires first, a generalization of identification results for variational autoencoders, the followed by second, a causal discovery subroutine, and third, learning an invariant predictor using the discovered causes. The procedure reads sound, and the results on common benchmarks look good, though I do not know how practical the approach would be in general.